# Rearrangement of the transmembrane domain interfaces associated with the activation of a GPCR hetero-oligomer

Li Xue [1,4], Qian Sun [1,4], Han Zhao[1,4], Xavier Rovira[2,3,4], Siyu Gai[1], Qianwen He[1], Jean-Philippe Pin [2], Jianfeng Liu[1] & Philippe Rondard [2]

G protein-coupled receptors (GPCRs) can integrate extracellular signals via allosteric interactions within dimers and higher-order oligomers. However, the structural bases of these interactions remain unclear. Here, we use the GABA$_B$ receptor heterodimer as a model as it forms large complexes in the brain. It is subjected to genetic mutations mainly affecting transmembrane 6 (TM6) and involved in human diseases. By cross-linking, we identify the transmembrane interfaces involved in GABA$_{B1}$-GABA$_{B2}$, as well as GABA$_{B1}$-GABA$_{B1}$ interactions. Our data are consistent with an oligomer made of a row of GABA$_{B1}$. We bring evidence that agonist activation induces a concerted rearrangement of the various interfaces. While the GB1-GB2 interface is proposed to involve TM5 in the inactive state, cross-linking of TM6s lead to constitutive activity. These data bring insight for our understanding of the allosteric interaction between GPCRs within oligomers.

[1] Cellular Signaling laboratory, International Research Center for Sensory Biology and Technology of MOST, Key Laboratory of Molecular Biophysics of MOE, and College of Life Science and Technology, Huazhong University of Science and Technology, Wuhan 430074 Hubei, China. [2] Institut de Génomique Fonctionnelle (IGF), CNRS, INSERM, Université de Montpellier, Montpellier 34094 Montpellier cedex 05, France. [3] Present address: Molecular Photopharmacology Research Group, The Tissue Repair and Regeneration Laboratory, University of Vic - Central University of Catalonia, C. de la Laura, 13, Vic 08500, Spain. [4] These authors contributed equally: Li Xue, Qian Sun, Han Zhao, Xavier Rovira. Correspondence and requests for materials should be addressed to J.-P.P. (email: jean-philippe.pin@igf.cnrs.fr) or to J.L. (email: jfliu@mail.hust.edu.cn)

G protein-coupled receptors (GPCRs) form the largest family of cell surface receptors and all cells are covered with dozens of different GPCR subtypes[1]. At the cellular level, multiple mechanisms have been identified that integrate the various GPCR-mediated signals. These mechanisms involve either cross-talk between signalling pathways[2], or allosteric interactions between receptors associated in dimers or higher-order oligomers[3–7]. Although largely debated[8,9], physical interactions between GPCRs allow either positive or negative cooperativity between protomers, both in homo-[3,7,10,11] and hetero-oligomers[5,12–17]. Recent studies highlight the potential role of such receptor assembly in physiopathological processes[14,18–20].

Numerous structural, biophysical and biochemical studies have investigated the quaternary organization of GPCRs[21–23]. However, the structural bases for GPCR assembly and allosteric interaction remain elusive. To date, the most compelling studies revealed the transmembrane helices TM4 and TM5 on one hand, and TM1 and TM7 on the other hand, form possible dimerization interfaces[20,24–27]. Surprisingly, the amplitude of the conformational changes associated with ligand occupancy is limited at these proposed interfaces. This limitation makes a possible allosteric control of one subunit by the other difficult. This lack of a clear view of the interfaces involved in GPCR allosteric interactions may be due to the dynamic interaction between receptor molecules, as revealed by single-molecule studies[24,28–30]. Elucidating how oligomers assemble and how the subunits functionally interact is key for our understanding of their possible physiological significance.

The GPCR for γ-aminobutyric acid (GABA), the GABA_B receptor, is involved in pre- and post-synaptic regulation of many synapses[31]. It is an excellent model to investigate the structural basis of cooperativity in higher-order oligomers for several reasons. (i) The functional unit is a mandatory heterodimer of two homologous subunits GABA_{B1} (GB1) and GABA_{B2} (GB2) (Fig. 1a)[32]. (ii) Allosteric interactions between the seven transmembrane helices (7TMs) of GB1 and GB2 lead to improved coupling efficacy of GB2[16]. (iii) GABA_B receptors have the propensity to form stable hetero-oligomers organized through interactions between the GB1 subunits[12,28,33–35] (Fig. 1b). (iv) Allosteric interactions between the heterodimeric units within such oligomers have been identified. These interactions allow a single heterodimer to bind ligand and activate G-proteins, within a tetrameric entity[12,35]. Despite this clear evidence of allosteric interactions between the subunits of the GABA_B oligomer, and the known structure of the active and inactive heterodimeric extracellular domain[36], little is known about 7TM structure.

Clarifying the structural bases of the allosteric interaction between GABA_B subunits is critical, as this receptor is an interesting target for the treatment of various diseases, including spasticity, pain and alcoholism[37]. Moreover, recent studies revealed the GABA_B receptor can be the target of auto-antibodies possibly at the origin of epilepsies and encephalitis[38]. In addition, mutations in the GABA_{B2} receptor gene have been recently reported to be associated with Rett syndrome and epileptic encephalopathies[39–41]. Most of them correspond to residues in the TM6 helix that could point out outside of the 7TM core (Fig. 1c), while one was found in TM3 buried of the middle of the 7TM core[40,41].

In this study, we reveal the 7TM domain interfaces in the GABA_B oligomers and we also document their dynamics during receptor activation. Our data are consistent with a concerted reorientation of the subunits associated with receptor activation. Altogether, these data provide important information on how GABA_B receptor oligomers are activated. Our data are more generally applicable to understanding the structural bases of the cooperativity observed in many GPCR dimers and higher-order oligomers.

## Results

**GB1 and GB2 constructs for cross-linking experiments**. In this study, our aim was to identify the various interfaces involved in interaction of the GABA_B receptor subunits in oligomers. For this, we decided to use cysteine cross-linking that gives a rather good resolution of the possible proximity between two residues in protein-protein interactions since it requires a distance below 8 Å between the Cβ of both cysteines. We was previously successfully

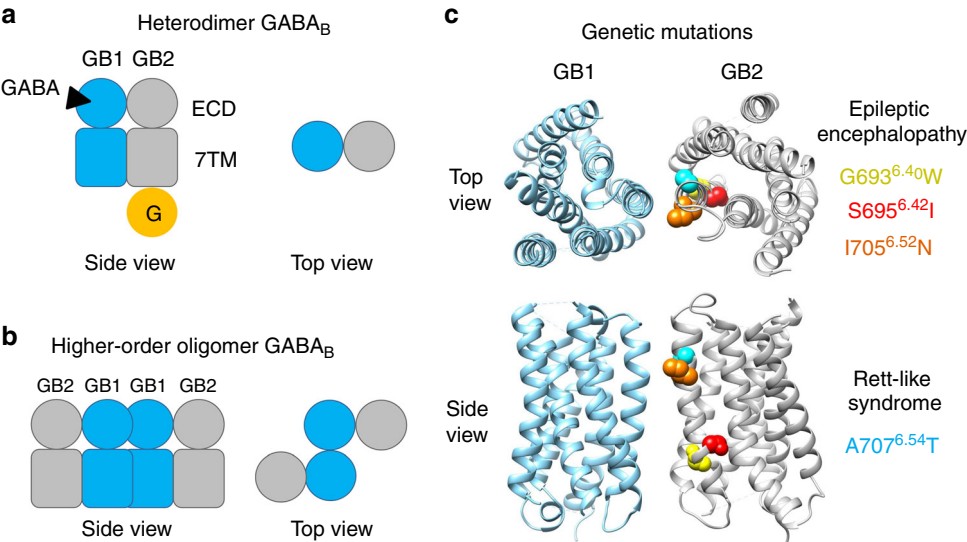

**Fig. 1** Schematic representation of the GABA_B receptor. **a** GABA_B forms an obligatory heterodimer made of the two subunits GABA_{B1} (GB1, blue) and GABA_{B2} (GB2, grey). GABA binds to the extracellular domain (ECD) of GB1, while the GB2 heptahelical domain (7TM) is responsible for G-protein activation. **b** GABA_B has the tendency to form stable higher-order hetero-oligomers that are likely organized through interactions between the GB1 subunits, while GB2 is likely not directly involved in these contacts. **c** Recently reported loss-of-function genetic mutations in GB2 7TM in human diseases. Most of these mutations affect residues in GB2^{TM6} (Gly693, yellow; Ser695, red; Ile705, orange; Ala707, cyan). These mutations produce a constitutively active receptor, except the mutation of Gly693 that has not been studied in functional assays

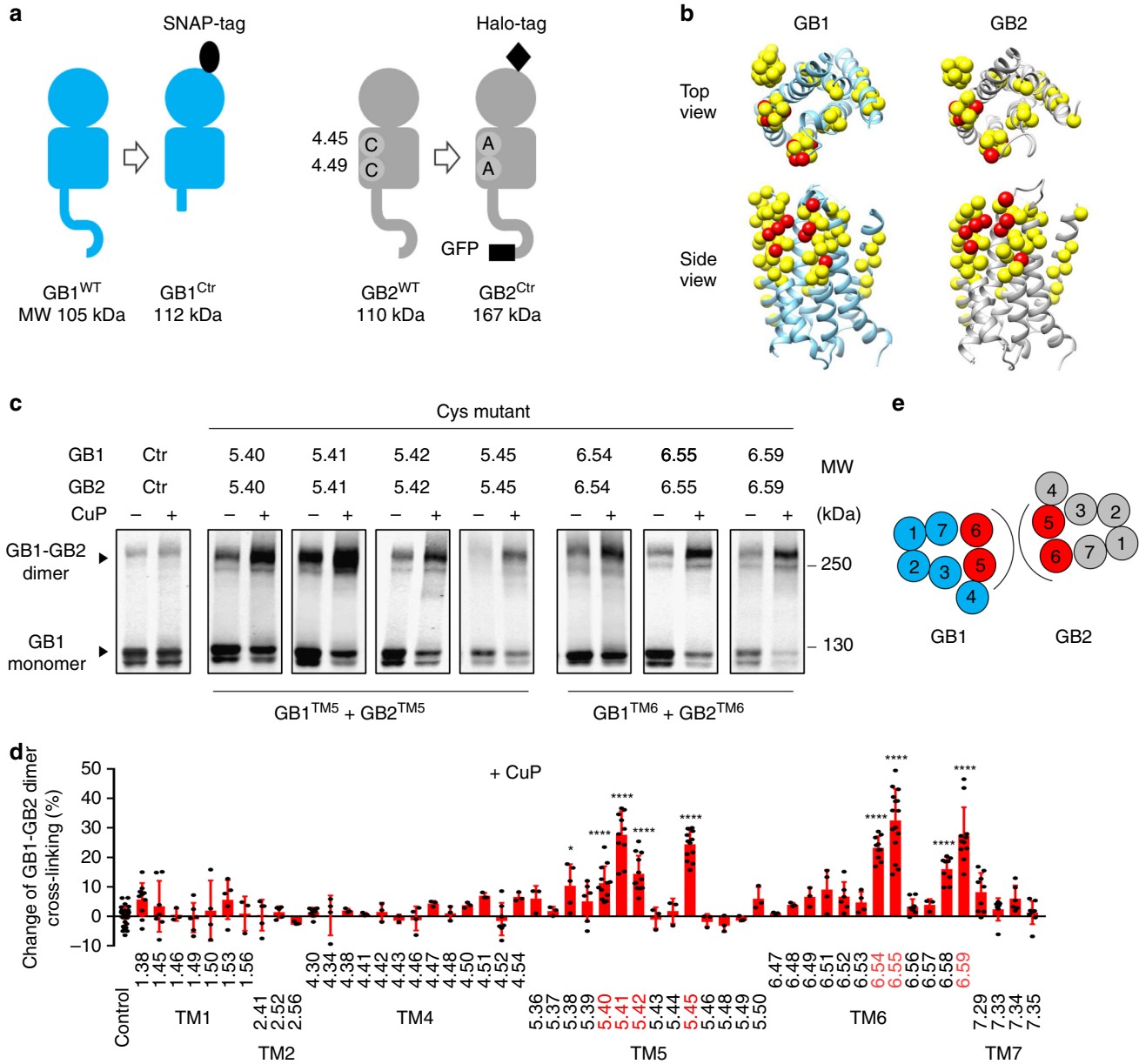

**Fig. 2** Cysteine cross-linking identifies TM5 and TM6 at the 7TM heterodimer interface. **a** Schematic representation of the GB1$^{Ctr}$ and GB2$^{Ctr}$ constructs used in the study. To easily distinguish GB1-GB2 and GB1-GB1 cross-linking in SDS-PAGE experiments, the molecular weight of the two subunits was modified. The SNAP-tagged full-length GB1 was truncated in the C-terminal region downstream of the coil-coiled region. Halo-tagged full-length GB2 was enlarged by adding a GFP tag at the C-terminal end of the subunit. To prevent the endogenous Cys producing unwanted disulphide bridges, the two indicated Cys residues in GB2$^{TM4}$ were changed to alanine. **b** 3D model of the 7TM of GB1 (blue) and GB2 (grey). All cysteine substitutions are highlighted by a yellow ball (α carbon), and those that cross-linked well in TM5 and TM6 (see panel c) by a red ball. **c** Cross-linking of the indicated cell surface SNAP-GB1 subunits labelled with fluorescent SNAP substrates, after treatment (+) or without treatment (−) with CuP. After SDS-PAGE in non-reducing conditions, GB1 monomers and GB1-GB2 dimers were detected via the fluorophore covalently attached to the receptors. MW, molecular weight. Data are representative of a typical experiment performed three times. **d** Change of GB1-GB2 dimer rate induced by CuP treatment for the "Control" heterodimer (GB1$^{Ctr}$ co-expressed with GB2$^{Ctr}$) and every indicated mutant (both GB1 and GB2 subunits having a Cys residue in the same position). Positions with a significant change were highlighted in red. Data are mean ± SD from at least three independent experiments ($n = 3$–6). Unpaired t test with Welch's correction with ****$P < 0.0001$ and ***$P < 0.001$, the other data being not significant. **e** Dimerization interface based on the results of the cross-linking experiments in the absence of ligand. TMs that can cross-link between GB1 and GB2 are highlighted in red

used this approach to study the metabotropic glutamate receptor type 2 (mGlu2)[42], that belong to the class C GPCRs as the GABA$_B$ receptor[43]. We used N-terminally SNAP-tagged GB1 and Halo-tagged GB2 (Fig. 2a) because they can be selectively and covalently labelled with non-cell permeant fluorescent substrates. Accordingly, only cell surface proteins are labelled, such that any oligomers retained in the intracellular compartment will not be

detectable[34,44]. This is especially important in the case of the heterodimeric GABA$_B$ receptor for which one subunit (GB2) is required for the other (GB1) to reach the cell surface. Indeed, GB1 non associated with GB2 is retained in intracellular compartments[45]. With this approach both subunits can easily be detected by their fluorescence after SDS-PAGE in non-reducing conditions and protein-transfer to membranes, without the need

of antibody labelling. SNAP-GB1 and Halo-GB2 have very similar molecular weights making distinguishing them difficult (Fig. 2a). Therefore, we shortened the C-terminal end of GB1 in our constructs and enlarged the C-terminal end of GB2 by adding a GFP tag. This gave easily distinguishable GB1 and GB2 subunits of 112 and 167 kDa respectively (Supplementary Fig. 1a, b). Accordingly, the GB1-GB2 heterodimers (279 kDa) can easily be separated from the GB1-GB1 dimer (224 kDa) by non-reducing SDS-PAGE (Supplementary Fig. 1c). To prevent unwanted disulphide bridges, we mutated the cysteines of GB2 TM4 (Cys609[4.45] and Cys613[4.49]; see nomenclature of the class C GPCR 7TMs[46]) to alanine (Supplementary Fig. 2a). These constructs are named 'control subunits' and referred to GB1[Ctr] and GB2[Ctr] in this study (Fig. 2a). Finally, we verified that these two engineered subunits have similar cell surface targeting and functional properties to wild-types (Supplementary Fig. 2b, c).

**Characterization of the GB1-GB2 7TM dimer interface**. To characterize the GB1-GB2 interface, we examined inter-subunit cross-linking between GB1[Ctr] and GB2[Ctr] carrying one cysteine residue in all the TMs, except TM3 that is mainly buried into the 7TM domain (Fig. 2b–d; Supplementary Figs. 3a, b, 4, 5 and 6). Only symmetric dimer interfaces were considered since the GABA_B receptor ECD is symmetric[36]. Asymmetric interfaces have been less described in the GPCR family, and they are all computational studies[24]. Therefore, only GB1 and GB2 with a cysteine at the same position were co-expressed.

One needs to be cautious in interpreting the cross-linking results with membrane proteins from the blots analysis. A background for the dimer band is observed in most samples and is enhanced by the introduction of cysteines in many locations. It is probably due to non-specific cross-linking or non-specific association of the subunits upon denaturation. Non-specific cross-linking could occur at the cell surface spontaneously or during treatment with oxidative copper-phenanthrolin (CuP) before stopping the cross-linking reaction with the alkylating agent N-ethylmaleimide. Alternatively, Cys-crosslinking can occur after protein denaturation due to the exposure of buried Cys. Indeed, GB1[Ctr] and GB2[Ctr] retain some reactive cysteine residues that could form a spontaneous or CuP-induced disulphide bridge, though with low efficiency. In addition, non-specific association is expected to occur upon membrane protein denaturation, especially if the proteins are already associated in the plasma membrane, due to hydrophobic interactions between the unfolded protein chains. GB1[Ctr] and GB2[Ctr] retain the coiled-coil domain existing in the C-terminal region of the GABA_B receptor that can favour SDS-resistant dimers not necessarily covalently linked[12,47], although they have not been observed by others[48]. In agreement, under basal conditions, a high variability in the ratio of GB1-GB2 dimer over the total of GB1 subunit is measured in the different experiments (Supplementary Fig. 4). This probably results from differences in expression level and in sample preparation between the experiments. Of note, treatment with the reducing agent dithiothreitol (DTT) just before running the blots showed that a large part of the GB1-GB2 heterodimer band is resistant indicating than these dimers result from a non-specific protein association (Supplementary Fig. 3b). Such band is most probably made of SDS-resistant heterodimers that are not covalently linked through a disulphide bridge between the GB1 and GB2 subunits.

Then to analyze specific Cys cross-linking, we concentrated our effort in identifying Cys positions for which a strong CuP-induced cross-linking can be observed. CuP is used to promote Cys crosslinking[42,49] because spontaneous oxidation of the Cys residues located the plasma membrane is not efficient[49]. To determine the efficiency of cross-linking between the two subunits induced by CuP, we have quantified the change in the rate of GB1-GB2 dimers to the total quantity of GB1 subunit detected on blots (Fig. 2d). The results revealed efficient cross-linking of GB1 and GB2 when Cys were introduced in TM5 or TM6. No such cross-linking was observed when Cys were introduced in TM1, 2, 4 or 7. No significant CuP-induced cross-linking was observed between GB1[Ctr] and GB2[Ctr] in which no Cys was introduced. These data strongly suggest that TM5 and TM6 of both subunits constitute the GB1-GB2 dimer interface (Fig. 2e).

Finally, we have to be aware of another possible limitation of our cysteine cross-linking strategy that is the trapping of interactions that can be transient, and some of them not being functionally relevant. It could be due to constant conformational dynamics of the proteins and their movement in the biological sample, or a cross-linking that could occur during the sample preparation and experiments. In order to relate these interactions with functional properties of the receptor, we have performed these cross-linking experiments in presence of ligands known to stabilize the active or inactive conformations of the GABA_B receptor.

**GB1-GB2 interface changes upon receptor activation**. We have then tested the dynamics of this interface. We have quantified the agonist effects on cross-linking to all the sites of the 7TM domains where Cys were introduced, including in TM5 and TM6 (Fig. 3a–c and Supplementary Fig. 7a). In the presence of the agonist GABA, GB1-GB2 cross-linking between the two TM5s was largely decreased for two positions, indicating that the two TM5s are less close in the active state. However, inter-TM6 cross-linking was strongly increased for several positions, indicated the two TM6s are become closer during activation. Based on these data, we propose a model where the GB1-GB2 dimer interface switches from TM5–6 in the absence of ligand (basal or inactive state) to mainly TM6 in the active conformation (Fig. 3d).

We were not surprised to observe GB1-GB1 cross-linking, when using a GB1 subunit carrying a Cys residue, as it was known that the GABA_B receptor can associate into larger complexes likely through GB1-GB1 interaction[12,35] (Fig. 3a–b). However, consistent with our proposed model, there was a strong increase in GB1-GB1 dimers cross-linked through their TM5 upon agonist stimulation (Fig. 3a). In addition, the small amount of GB1-GB1 dimer cross-linked through their TM6 observed in the presence of the antagonist is no longer measured in the presence of the agonist (Fig. 3b). Of note, in these experiments the cross-linked bands were only partially decreased after DTT (Supplementary Fig. 7b), suggesting that even after reduction of the cross-linked disulphide bridges, none covalent SDS-resistant interactions remain between GB1 and GB2, as discussed above, or between two GB1 subunits[50].

Overall, these data indicate a dynamic interaction between the subunits in the GABA_B oligomer whereby GB1 TM6 switches from mainly contacting GB1 in the inactive state to contacting GB2 in the active state (Fig. 3d).

**Locking GB1-GB2 TM6 interface stabilizes an active state**. As our results suggest a TM6-TM6 interaction in the active state of the heterodimer, we postulated that this interface may be critical in the activation process since GB1 7TM strongly favours GB2 7TM coupling to G proteins[16]. We therefore cross-linked the TM6 domains in the heterodimer using the mutants GB1 I824C[6.59] and GB2 L711C[6.59] that had an efficient cross-linking between GB1 and GB2 at the TM6 level (Fig. 2d), but that could not be further increased by the agonist (Fig. 3c). Doing so, we

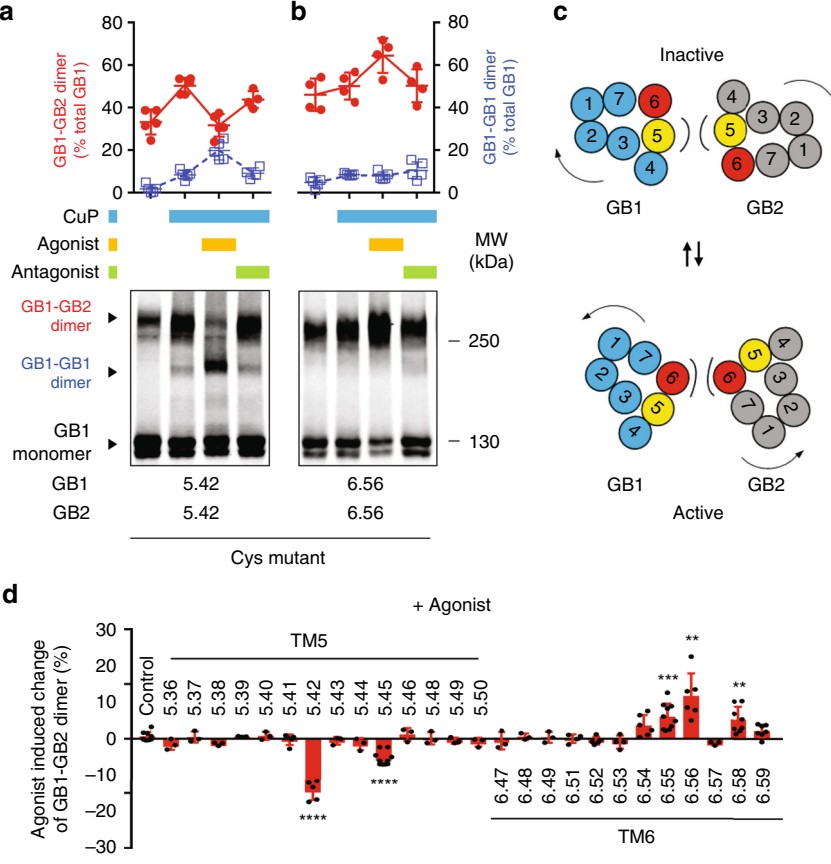

**Fig. 3** The interface of the heterodimer is switched from TM5 to TM6 during activation. **a, b** The cell surface SNAP-GB1 containing the indicated single cysteine substitution was cross-linked with the indicated GB2 cysteine mutant. The results were obtained for the symmetric interface TM5 and TM6, after pre-incubation with the agonist GABA or the competitive antagonist CGP54626 and with CuP. The percentage of GB1-GB2 heterodimers (in red) and GB1-GB1 homodimers (in blue) relative to the total amount of GB1 subunit was quantified by imaging the fluorescent blots. **c** Change of GB1-GB2 dimer rate induced by the agonist and determined by GB1-GB2 dimer quantification before and after GABA treatment. Data are mean ± SD from at least three independent experiments ($n = 3$–5). Unpaired t test with Welch's correction with ****$P < 0.0001$, ***$P < 0.001$ and **$P < 0.01$, the other data being not significant. GABA and CGP54626 were used at 100 μM. **d** Model highlighting the TMs involved in the dimerization of GB1-GB2 heterodimers in the inactive state (TM5, yellow) and in the active state (TM6, red)

observed a robust constitutive activity after CuP treatment in basal conditions (Fig. 4a). This constitutive activity was only slightly further stimulated by the full agonist GABA. This basal activity of the GABA$_B$ mutant correlated with the amount of receptor at the cell surface (Fig. 4b), and it cannot be blocked by the competitive antagonist (Fig. 4c). Importantly, CuP treatment itself had no effect on the GABA$_B$ receptor activity (Supplementary Fig. 8a). In the absence of CuP treatment, these mutated GABA$_B$ constructs had a similar activity than the wild-type (Supplementary Fig. 8b, c). Conversely, when the putative inactive interface was stabilized by cross-linking GB1$^{TM6}$ with GB2$^{TM4}$ (Fig. 4d), using the mutants GB1 I824C$^{6.59}$ and GB2 A616C$^{4.52}$ that cross-linked well (Fig. 4e), the activation of the receptor by agonist was impaired (Fig. 4f). This activation is not completely suppressed likely because only a fraction of the receptors are cross-linked. Of note, the activation of the receptor by agonist was not impaired by the reversed pair GB1$^{4.52}$ with GB2$^{6.59}$, and the GB1$^{5.42}$ with GB2$^{5.42}$ cross-linking (Supplementary Fig. 8d, e). It is probably because in these cross-linking experiments the oligomer is stabilized in a conformation closer to the active state by GB1-GB1 cross-linking through two GB1$^{TM4}$ and two GB1$^{TM5}$, respectively (see below). Accordingly, the GB1-GB1 dimer rate is strongly increased by the agonist in the GB1$^{4.52}$ with GB2$^{6.59}$ (Supplementary Fig. 8e) and GB1$^{5.42}$ with GB2$^{5.42}$ (Fig. 3a).

**Model of the rearrangement at the 7TM heterodimer interface.** Based on the above experimental data, we propose a 3D model for the activation of the GABA$_B$ receptor, where in the inactive state, the heterodimer interface would be formed mainly by the two TM5s, plus GB1$^{TM6}$ and GB2$^{TM4}$ (Fig. 5a). During activation, a rearrangement of this interface would occur such that in the active state, the interface mainly involves the TM6s of both GB1 and GB2, as recently proposed in mGlu receptors[42,51]. Of note, our previous experimental data have shown a higher probability to cross-link TM4s in mGlu2 homodimers[42], than in the GABA$_B$ heterodimer in this study. Indeed, we did not obtain any specific cross-linking between GB1-TM4 and GB2-TM4 in the resting (Supplementary Fig. 3b) and active state of the receptor (Supplementary Fig. 7a). We then propose that the amplitude of the relative reorientation between the 7TM dimer appears smaller in GB1-GB2 than in the mGlu2 homodimer (Fig. 5b). Our proposal is consistent with the observation of a smaller conformational change of the GABA$_B$ ECD compared to mGluR ECD, as previously reported based on crystal structures and FRET experiments[36,52].

**GB1 7TM interaction in the oligomer during activation.** As observed above, GB1 mutants can be cross-linked not only with GB2 but also with themselves. It is consistent with the ability of

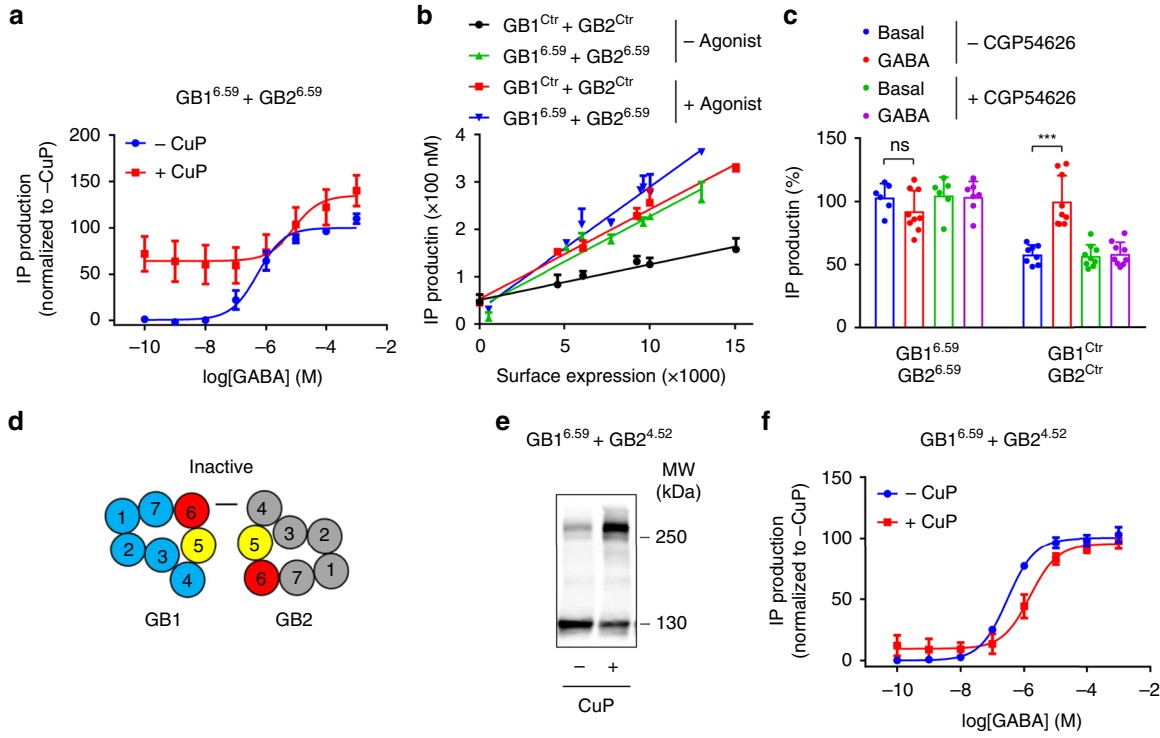

**Fig. 4** Disulfide cross-linking confirms the GB1-GB2 TM6 active interface and the resting interface. **a** Inositol phosphate (IP) production in cells that co-express the mutants GB1[6.59] and GB2[6.59] after treatment with or without CuP, and stimulation with GABA. Results are mean ± SD from three independent experiments performed in triplicates. **b** In both the control receptor (after stimulation with GABA) and the co-expressed mutants GB1[6.59] and GB2[6.59], IP production is proportional to the amount of SNAP-tagged GB1 at the cell surface, as measured by fluorescence after labelling with SNAP-Red substrate and then treatment with CuP. GABA was used at 100 μM. Data are mean ± SD from a typical experiment performed three times. **c** Treatment with the indicated competitive antagonist does not reverse the constitutive activity after GB1-GB2 TM6s cross-linking. GABA and CGP54626 were used at 1 and 10 μM, respectively. Data are mean ± SEM from a typical experiment performed three times. Unpaired t test with Welch's correction with *$P < 0.1$, *ns*, not significant. **d-f** Stabilizing the inactive GB1-GB2 interface (**d**) by co-expressing the indicated mutants that cross-link well upon CuP treatment (**e**), impairs IP accumulation induced by GABA (**f**). Data are mean ± SD from a typical experiment performed three times

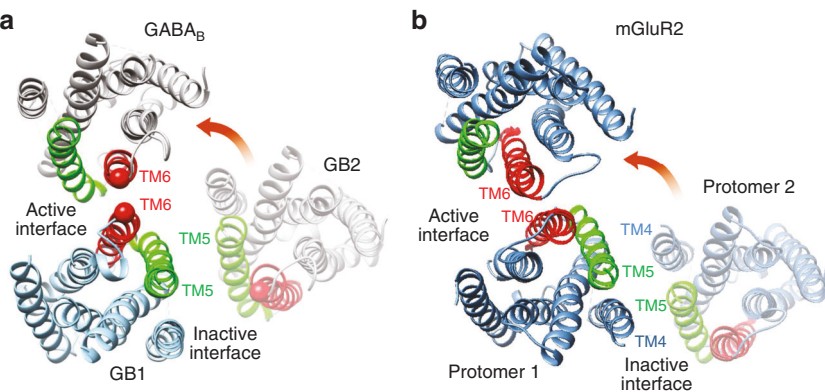

**Fig. 5** Rearrangement of the transmembrane domain interface during GABA_B heterodimer activation 3D model of the GB1-GB2 7TM heterodimer (**a**) and mGluR2 7TM homodimer (**b**) in the resting and active orientations. Based on these models, the amplitude of the relative reorientation of two 7TMs in the dimer might be smaller in the GABA_B receptor than in mGluR2

GABA_B receptors to form large complexes through GB1-GB1 interaction[12,34]. In order to identify the GB1 interfaces involved in the formation of oligomers, we performed GB1-GB1 cross-linking in conditions where we would not have GB1-GB2 cross-linking. Therefore, we then examined the possible cross-linking between GB1[Ctr] subunits carrying one Cys residue in various TMs, co-expressed with GB2[Ctr] that do not contain introduced Cys (Fig. 6a–b).

Under basal conditions, CuP treatment resulted in a strong increase of GB1-GB1 cross-linked dimers for the cysteine mutant

in TM4 and TM6 (Fig. 6c), where a single Cys mutation was introduced either in GB1[TM4] or in GB1[TM6]. In the same conditions, CuP treatment increased GB1-GB1 cross-linked dimers to a lower extent for TM1, TM5 and TM7 (Fig. 6d). These results suggest there are higher-order oligomers in the inactive state, where one GB1 subunit forms two different interfaces with two other GB1s, one mediated by TM4 and the other by TM6 (Fig. 6e). This model is also consistent with the GB1[TM5]-GB2[TM5] interface we proposed for the GABA_B heterodimer in the inactive state, where both TM5s are buried in the

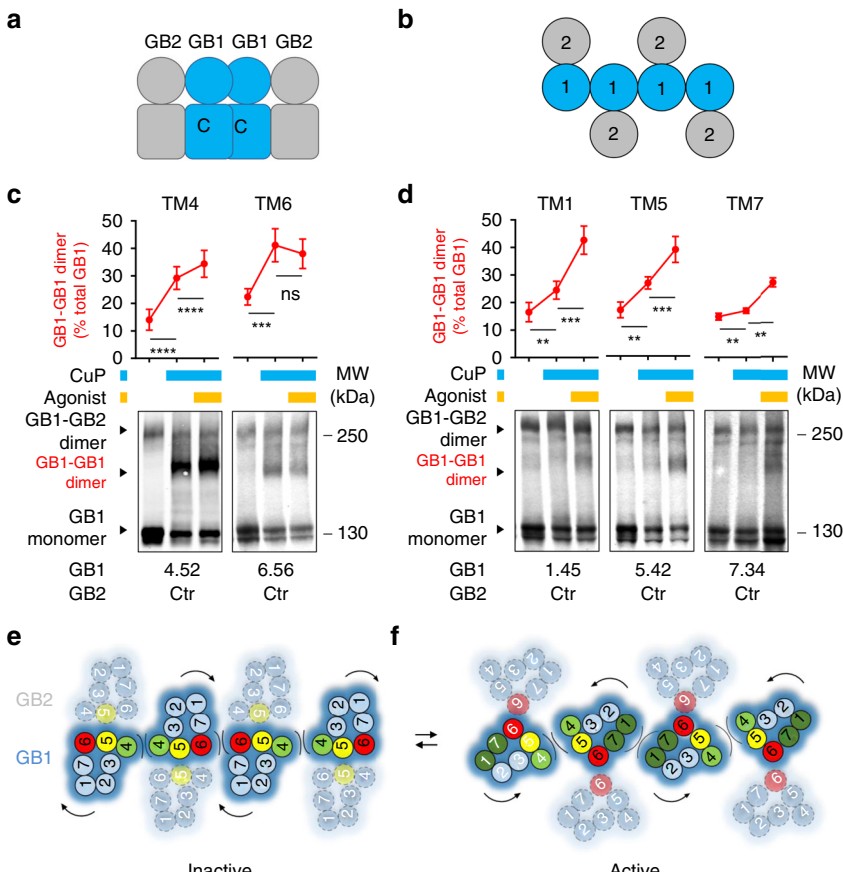

**Fig. 6** Interactions between GB1 7TMs in GABA$_B$ oligomers during activation. **a**, **b** Schematic representation of a GABA$_B$ oligomer in lateral (**a**) and top view (**b**). **c**, **d** Blots showing cross-linking of cell surface SNAP-GB1 subunits containing a single cysteine substitution in TM1, TM4, TM5, TM6 or TM7, with GB2$^{Ctr}$ after pre-incubation or not with GABA (agonist) and with CuP, as indicated. The percentage of GB1-GB1 homodimers (in red) relative to the total amount of GB1 subunit was quantified from the fluorescent images. Data are mean ± SD from three independent experiments. Paired t test with Welch's correction with ****$P < 0.0001$, ***$P < 0.001$ and **$P < 0.01$, or not significant (ns). **e**, **f** Model for the structural organization of the GABA$_B$ 7TMs in higher-order oligomers in the inactive and active state. Interfaces at the GB1 subunits are highlighted

interface of the heterodimer (Fig. 5a), then the probability of two GB1$^{TM5}$ being crosslinked in the inactive state is low.

In contrast, a strong increase of the GB1-GB1 cross-linking was induced by agonist for Cys located in TM1, TM5 and TM7 (Fig. 6d), but to a lower extent for TM4 while no significant change was obtained for TM6 (Fig. 6c). These results indicate that two main interfaces are formed between the GB1 subunits in the higher-order oligomers during activation, one being TM5 and the other TM1-TM7 interface (Fig. 6f). This active state of the oligomers is consistent with the movement of GB1$^{TM6}$ that switches to the GB2 interface during activation (Fig. 5a). Such reorientation of GB1$^{TM6}$ should limit its exposure to form cross-linking with another GB1$^{TM6}$, consistent with no increase in cross-linking between two GB1$^{TM6}$ upon agonist treatment (Fig. 6c). Of note, in these experiments the GB1-GB1 cross-linked bands were only partly sensitive to DTT (Supplementary Fig. 9), suggesting that even after reduction of the cross-linked disulphide bridges, none covalent but strong interactions remain between GB1 subunits, as stated above.

**Model of the two interfaces between GB1s in oligomers.** To further support this oligomerization model and validate which GB1-GB1 interfaces are made at a given time, we measured the high-molecular weight species formed by the cross-linked GB1 subunits. We explored which pairs of cysteines introduced

in the GB1 7TM cause higher-order oligomers, when co-expressed with a non-mutated GB2 (Fig. 7a; Supplementary Fig. 10a). These high-molecular-weight complexes only formed for those mutants of GB1 that can form one interface through two GB1$^{TM4}$ or GB1$^{TM5}$ and another interface between two GB1$^{TM1}$, GB1$^{TM6}$ or GB1$^{TM7}$ (Fig. 7b). There were no high-molecular-weight complexes with GB1$^{Ctr}$ co-expressed with GB2$^{Ctr}$, and also with most of GB1 double mutants co-expressed with GB2$^{Ctr}$ (Fig. 7a; Supplementary Fig. 10a). These oligomers are consistent with the cross-linking of at least three GB1 subunits through two different interfaces of GB1 in the inactive state, one mediated by TM4s or/and TM5s and the other by TM1s, TM6s or TM7s (Fig. 7c). Of note, in these experiments the oligomer cross-linked bands were sensitive to DTT (Supplementary Fig. 10b) although that not totally, suggesting none covalent but strong interactions remain between GB1 subunits, as stated above.

Interestingly, receptor activation increased the intensity of the oligomeric band when the symmetric GB1$^{TM4}$ interface was cross-linked together with GB1$^{TM1}$ or GB1$^{TM7}$ interface (Fig. 7a). In addition, the symmetric GB1$^{TM5}$ interface was cross-linked together with GB1$^{TM1}$. These results are consistent with the active state of the oligomers proposed above (Fig. 6f). Of note, our data suggested that a simultaneous cross-linking of the two interfaces mediated by TM5s and TM7s within the same GB1 subunit to form oligomers is not possible. Indeed, a double mutant of GB1 carrying one cysteine in TM5 (I771C$^{5.42}$) and one in TM7

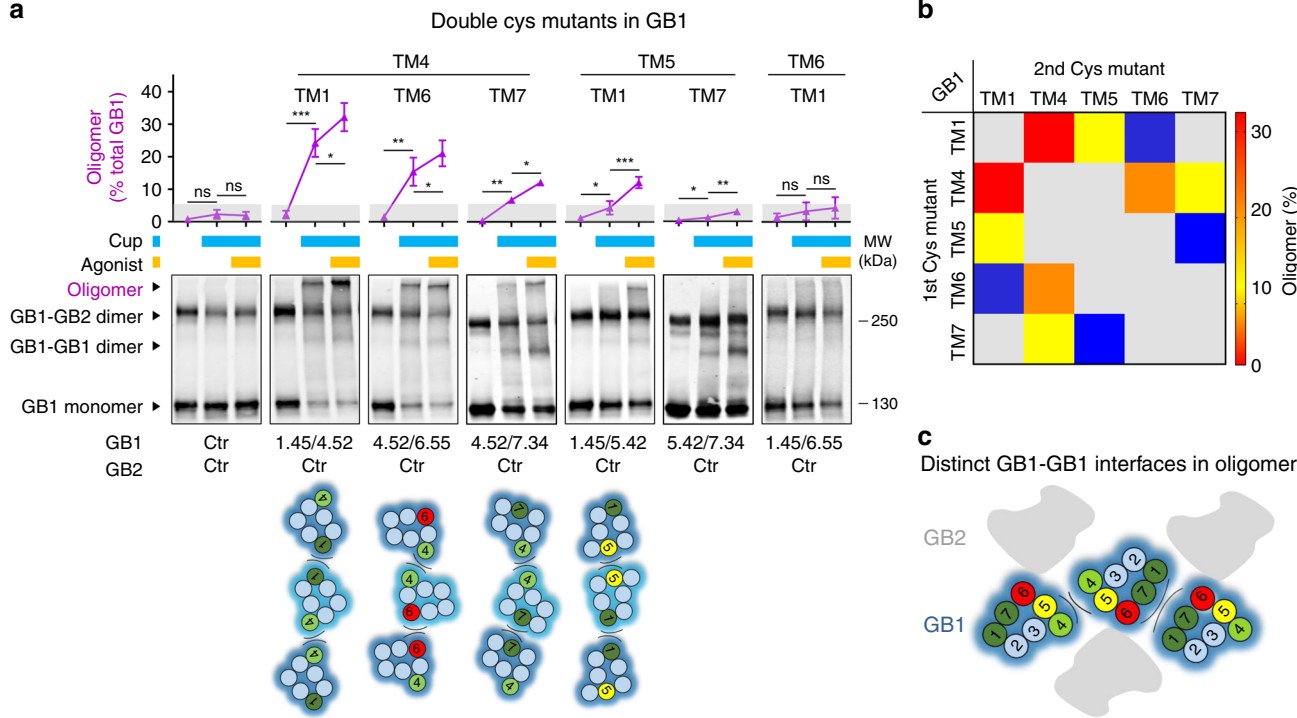

**Fig. 7** High-molecular-weight complexes confirm the two interfaces between GB1s in oligomers. **a** Quantification of the oligomers obtained after cross-linking of the double cysteine substitution in different TMs of the GB1 subunit, after pre-incubation or not with GABA and with CuP, as indicated. The percentage of oligomers (in purple) relative to the total amount of GB1 subunit was quantified from the fluorescent blots. The pictograms indicate the possible cross-linking of three GB1 subunits that could form the oligomer band of the corresponding blot. These schemes are from snapshots of the GABA$_B$ oligomer 3D model when morphing are performed between the inactive and active states (see Figs. 9a, b). GABA was used at 100 µM. Data are mean ± SD of at least three individual experiments (n = 3–5). Paired t test with Welch's correction with ***$P < 0.001$, **$P < 0.01$ and *$P < 0.1$, the other data being not significant (ns). **b** Quantification of the oligomers (% of total GB1) obtained for the indicated pairs of cysteines in panel (**a**), after cross-linking in presence of CuP and GABA. **c** Model of the 7TM of GABA$_B$ oligomers highlighting the two distinct and possible interfaces between the GB1 subunits during activation

(L838C$^{7.34}$) produced no oligomer but the GB1-GB1 dimer rate that was further increased by GABA (Fig. 7a). In contrast the simultaneous GB1 two interfaces TM5s/TM1s, TM4s/TM1s or TM4s/TM7s are possible.

**A disease-causing mutation stabilizes the active interfaces.** We have introduced the genetic mutation S694I$^{6.42}$ in our rat GB2 constructs (equivalent to genetic mutation S695I$^{6.42}$ in human GB2), that produced a strong constitutive activity of the the GABA$_B$ receptor (Supplementary Fig. 11), and as recently reported[40]. In the absence of agonist, this mutation stabilized the active interface of the heterodimer unit mediated by both TM6s as measured by the increased GB1-GB2 cross-linked upon CuP treatment (Fig. 8a–b). In addition, this mutation stabilized the active interface between the GB1 subunits in the oligomer in the basal state, as measured by a strong crosslinking between the GB1 TM5s upon CuP treatment (Fig. 8c–d). Altogether these data are consistent with a constitutive activity of the receptor induced by this mutation. This later is also associated with the stabilization of an oligomer organized in an active assembly.

**Model of the active and inactive 7TM oligomer interfaces.** Altogether, on the basis of the cysteine cross-linking results, we propose a 3D model of the 7TM oligomer using four molecules of heterodimers, named *A-D* (Fig. 9). In the resting state, one heterodimer interacts with two others through the GB1 subunits, through two symmetric interfaces mediated by GB1$^{TM4}$ and GB1$^{TM6}$ that are on the opposite face of GB1 (Fig. 9a).

Accordingly, GB1$^{TM4}$ of the heterodimer *B* interacts with GB1$^{TM4}$ of the heterodimer *C*, while GB1$^{TM6}$ of the heterodimer *B* interacts with the GB1$^{TM6}$ of the heterodimer *A*. In the active state, two new interfaces are formed: (i) a GB1 interface TM4-TM5 made by the heterodimers *B* and *C*; (ii) a GB1 interface TM1-TM7 between the heterodimers *A* and *B* (Fig. 9b). Our model is compatible with the reorientation of the TM5s and TM6s at the interface between GB1 and GB2 during activation, as proposed above (Fig. 5a). Finally, this active state of the oligomer allows the coupling of one G-protein by dimer (Fig. 9c).

## Discussion

The GABA$_B$ receptor was the first clear example of a mandatory heterodimeric GPCR[53], and this discovery stimulated research on the putative dimerization of other GPCRs. Furthermore, the GABA$_B$ receptor was more recently shown to associate into larger complexes made of two or more heterodimers[12,28,34], and this was confirmed in native tissues in several ways[12,54,55]. However, the structural bases of the interactions are still unclear. Here, using Cys cross-linking experiments, we propose a model for the GABA$_B$ 7TM assembly within a GABA$_B$ oligomer, involving dynamic and concerted movements between the subunits associated with receptor activation. Interestingly, we identified TM6, the TM known to undergo major conformational change upon GPCR activation[56,57], to switch interfaces. Furthermore, we demonstrate that a TM6-TM6 interaction between GB1 and GB2 is sufficient for receptor activation.

We propose an organization of the GABA$_B$ higher-order oligomers in rows at the surface of live cells. Within these oligomers,

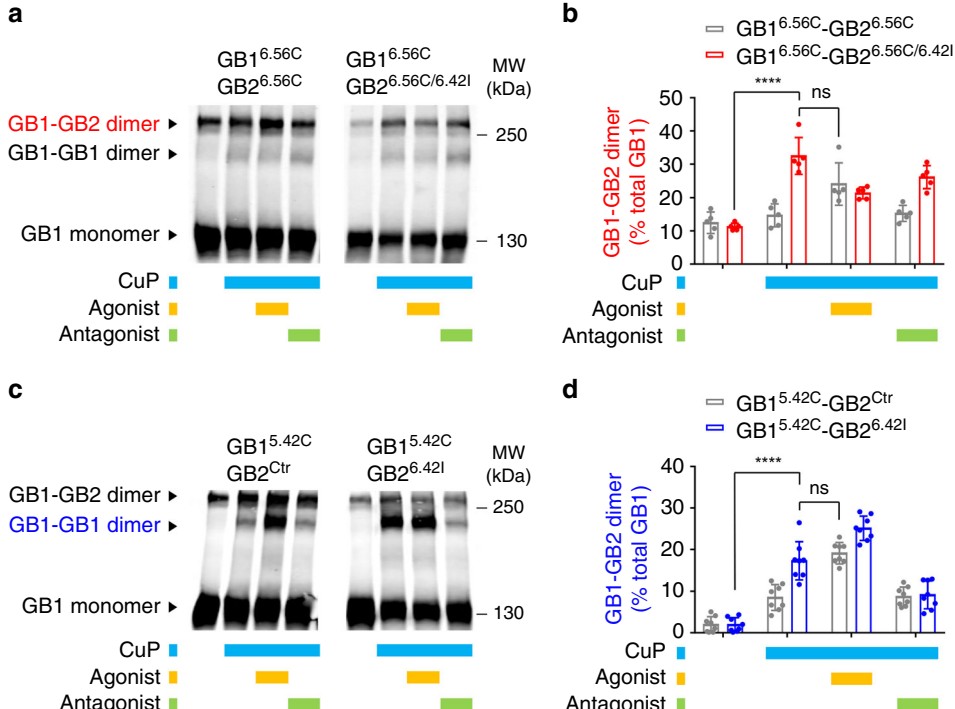

**Fig. 8** A disease-causing mutation stabilizes the active interface of the dimer and oligomer. **a**, **b** Quantification of the GB1-GB2 cross-linking for the GB1[6.56] and GB2[6.56] cysteine mutants containing or not the genetic mutation S695I[6.42] in the GB2 subunit, in the indicated conditions and as described in Fig. 3. Both cysteine mutation and the genetic mutation have been introduced in the rat GB1[Ctr] and GB2[Ctr]. **c**, **d** Quantification of the GB1-GB1 cross-linking for the GB1[5.42] single cysteine mutant co-expressed with GB2[Ctr] containing or not the genetic mutation S695I[6.42]. GABA and CGP54626 were used at 100 μM. Data are mean ± SD from at least three independent experiments (*n* = 3–5). Unpaired *t* test with Welch's correction with ****$P < 0.0001$, or not significant (ns)

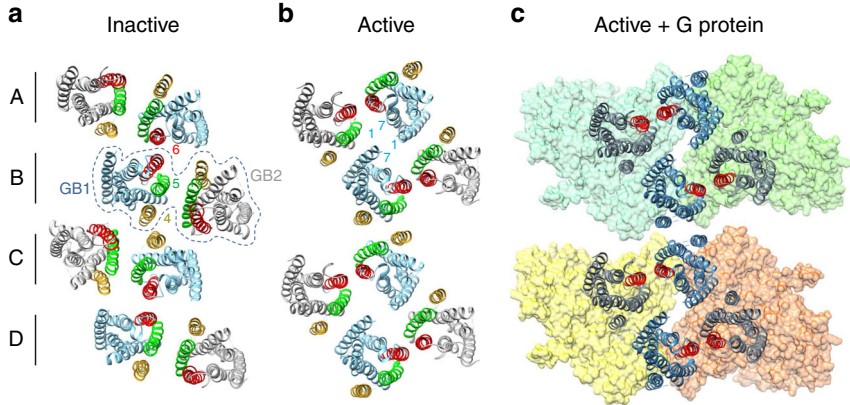

**Fig. 9** Agonist-induced rearrangement of the 7TMs in the GABA_B oligomer during activation. **a**, **b** 3D model of the 7TM oligomer in the inactive and active orientations. The dashed line highlights a minimal functional receptor made of GB1 and GB2 (heterodimer *B*). Heterodimer *A* is proposed to assemble the tetramer with *B*, and *C* to form oligomer with the tetramer *A*-*B*. In this model, stabilization of the tetramer interactions is made by the symmetric interfaces with GB1[TM6] in the resting state, and with GB1[TM1−TM7] in the active state. Stabilization of the oligomer would be through the symmetric interfaces with GB1[TM4] in the resting state and with GB1[TM4−TM5] in the active state. TM4, TM5 and TM6 of GB1 and GB2 are in yellow, green and red, respectively. **c** Model of the active oligomer coupled to four Gαβγ proteins based on the structure of the complex between the active β2-adrenergic receptor and the G protein previously reported[68]

GB1 subunits are assembled in lines via two opposite sides of their 7TMs, while the GB2 subunits are on the side. This model is supported by the large FRET signal previous reported between GB1 subunits, and a quasi-absence of FRET between GB2 N termini[12,34], even though GB2 subunits co-diffuse at the neuronal surface indicating they are in the same receptor complex[54]. This organization may also explain the observed ordered arrays of GABA_B receptors in transfected cells[28]. In class A GPCRs, similar rows have also been proposed for the organization of rhodopsin[21,22], a structure that could be destabilized by genetic mutations at the TM1 and TM5 interfaces then leading to retinitis pigmentosa[20]. Our data suggest that rows of GABA_B receptors may form spontaneously through two distinct GB1 interfaces, TM4–5 and TM1–7. This GABA_B receptor organization is

consistent with the proposed interfaces involved in many class A GPCR oligomerization[23,49,58], for which both TM4–5 and TM1–7 were the most frequently proposed[24,26].

In the $GABA_B$ oligomers, we propose that a dimer of dimers can form a minimal repeat unit. This tetramer is stabilized by interactions between the two GB1 subunits through their symmetric TM1-TM7 interface in the active state (Fig. 9b). This model is supported by the organization of the $GABA_B$ ECD in a tetramer, stabilized by interactions between the lobes 2 (lower lobes) of two GB1 VFTs[12]. As a consequence of this tetramer organization, the higher-order oligomers would be stabilized by the symmetric $GB1^{TM4}$ interface between two tetramers. Interestingly, when active G-protein is added to the receptor in our 3D model, the G-protein interacts with two GB1 subunits within the same tetramer. Most important, it is even possible for two G-proteins to couple to one $GABA_B$ tetramer (Fig. 9c). Thus the hypothesis that only one G-protein is activated by a tetramer[12] could not be explained by structural steric reasons at the level of the 7TMs. Instead, it could be due to the negative allosteric between two heterodimers within a tetramer, as recently reported for the $GABA_B$ ECDs[35].

In our model, the $GABA_B$ tetramer has a rhomboid shape structure. Rhomboids that has been proposed for several class A GPCRs that form spontaneous tetramers[23,58]. In addition, a rhomboid organization for the tetramer could explain the smaller amplitude of the relative reorientation in $GABA_B$ 7TM heterodimer compared to the 7TM of mGlu2 dimers that do not form constitutive oligomers[34,59,60]. This small rearrangement between the two 7TMs in the $GABA_B$ receptor heterodimer is also consistent with the limited conformational changes between the active and inactive states at the level of the ECDs[36], and the negative allostery between the two heterodimers[35].

We demonstrate here that the GB1-GB2 heterodimer is the minimal functional unit within tetramers. This is best illustrated by the receptor full constitutive activity resulting from $GB1^{TM6}$-$GB2^{TM6}$ crosslinking. A key determinant of a tetramer is the TM6 of GB1, that binds another GB1, in the inactive state, but binds GB2 in the active state. This concerted rearrangement of the various interfaces of GB1 during activation could be responsible for the positive cooperativity between the two 7TMs in the heterodimer. Indeed, we have previously demonstrated that the GB1 7TM activation is critical for stabilizing the active state for GB2 activation[16].

The switching of $GB1^{TM6}$ from one interface to another during activation could be also responsible for the asymmetric activation of the two 7TMs in the heterodimer. Indeed, both $GB1^{TM6}$ and the G-protein could be responsible for allowing a single 7TM domain in a heterodimer to reach a conformation compatible with G-protein activation[32]. Similarly in the homodimeric and heterodimeric mGluRs, one TM6 in the dimer could also be responsible for the asymmetric functioning of the 7TMs, where only one subunit of the dimer couples to the G-protein[15]. Indeed, in the mGlu2–4 and $GABA_B$ heterodimers, the G-protein is only activated by one of the subunits, namely mGlu4 and GB2, respectively. In these heterodimers, functional asymmetry is not due to the fact that it is only the G-protein-bound subunit that can change its conformation. Indeed, the associated subunit also reaches a specific conformation that positively acts on the G-protein-activating subunit[15,16]. The asymmetric functioning of TM6, as indicated by our data on $GABA_B$ and mGlu receptors, likely explains the allosteric interaction within class A GPCR dimers. Indeed, in many cases, a negative allosteric interaction has been reported, with one subunit only being able to reach a G-protein activating state[3,61].

The dynamic changes we observe at TM6 help to explain the many disease mutations there[39–41]. Interestingly, several of these

mutations in $GB2^{TM6}$ including those localized near the extracellular part of TM6 produce a $GABA_B$ receptor that is constitutively active, suggesting the mutations favour $GB1^{TM6}$-$GB2^{TM6}$ interactions, as demonstrated by one of them in the present study. Finally, auto-antibodies against the GB1 ECD were identified in a number of patients with encephalitis leading to loss of function of $GABA_B$ receptor[38,62]. The large and concerted movement proposed during activation of the $GABA_B$ oligomer offers multiple inroads for these antibodies to affect $GABA_B$ function.

In summary, we provide a model of dynamic interaction between 7TM protein subunits in a well-recognized oligomer, and we propose a key role for TM6 in this process. Although our model starts to explain allosteric interaction between GPCRs, these findings may be specific for the $GABA_B$ receptor, and other class C GPCRs or all GPCRs. These data provide the steps and future studies will determine the general applicability of the structural organization and allostery to GPCR dynamics.

## Methods

**Materials**. GABA (γ-aminobutyric acid) and dichloro(1,10-phenanthroline)copper (II) were purchased from Sigma-Aldrich (St. Louis, MO, USA). CGP54626 was from Tocris Bioscience (Ellisville, MO, USA). Lipofectamine 2000 and Fluo4-AM were obtained from Life Technologies (Carlsbad, CA, USA). SNAP-Surface® Alexa Fluor® 647 was from New England Biolabs, whereas HaloTag® Alexa Fluor® 660 was from Promega (Beijing) Biotech Co., Ltd.

**Plasmids and transfection**. The pRK5 plasmids encodes either the wild-type rat GB1a, tagged with HA and SNAP inserted just after the signal or the wild-type rat GB2 tagged with Flag and Halo inserted just after the signal peptide (Supplementary Fig. 12). $GB1^{Ctr}$ was obtained from rat GB1a wild-type sequence by deleting the last 32 amino acids encoding for GB1. $GB2^{Ctr}$ was obtained from rat GB2 wild-type sequence by adding a GFP-tag at the C-terminal end of GB2. The cysteine substitutions were generated by site-directed mutagenesis using the QuikChange mutagenesis protocol (Agilent Technologies) using the primers described in Supplementary Fig. 13 and Supplementary Fig. 14 for the GB1 and GB2 mutants, respectively.

HEK293 cells (ATCC, CRL-1573) were cultured in Dulbecco's modified Eagle's medium (DMEM) supplemented with 10% FBS and transfected by electroporation. Unless stated otherwise, $10^7$ cells were transfected with plasmid DNA containing the coding sequence of the receptor subunits, and completed to a total amount of 10 μg of plasmid DNA with the empty vector pRK5. For the determination of intracellular calcium measurements and inositol phosphate (IP) accumulation, the cells were also transfected with the chimeric G-protein $Gq_{i9}$, which allows the coupling of the recombinant $GABA_B$ receptor to the phospholipase C[52].

**Cross-linking and fluorescent-labeled blot experiments**. Forty-eight hours after electroporation, adherent HEK293 cells plated in 12-well plates were labeled with 100 nM SNAP-Green and 3.5 μM Halo-Red in culture medium at 37 °C for 1 h. Then, cells were incubated with drug (each at 100 μM) or PBS at 37 °C for 30 min. Afterwards, cross-link buffer (1.5 mM Cu(II)-(o-phenanthroline), 1 mM $CaCl_2$, 5 mM $Mg^{2+}$, 16.7 mM Tris, pH 8.0, 100 mM NaCl) was added at room temperature for 20 min. After incubation with 10 mM N-ethylmaleimide at 4 °C for 15 min to stop the cross-linking reaction, cells were lysed with lysis buffer (containing 50 mM Tris (pH 7.4), 150 mM NaCl, 1% NP-40, 0.5% sodium deoxycholate) at 4 °C for 1 h. After centrifugation at 12,000 × g for 30 min at 4 °C, supernatants were mixed with loading buffer at 37 °C for 10 min. In reducing conditions, samples were treated with 100 mM DTT in loading buffer for 10 min before loading the samples. Equal amounts of proteins were resolved by 29:1 acrylamide:bisacrylamide and 3–9% SDS-PAGE. For oligomer analysis, 59:1 acrylamide:bisacrylamide and 6% SDS-PAGE were used. Proteins were transferred to nitrocellulose membranes (Millipore). Membrane were imaged on an Odyssey CLx imager (LI-COR Bioscience, Lincoln, NE, USA) at 600 nm and 700 nm.

**Cell surface quantification**. Detection of the HA- and Flag-tagged constructs at the cell surface by ELISA was performed. Twenty-four hours after transfection, the HEK293 cells were fixed with 4% paraformaldehyde, blocked with 10% FBS. HA-tagged constructs were detected with a monoclonal rat anti-HA antibody 3F10 (Roche) at 0.5 μg/mL and goat anti-rat antibodies coupled to horseradish peroxidase (Jackson Immunoresearch, West Grove, PA) at 1.0 μg/mL. Flag-tagged constructs were detected with the mouse monoclonal anti-Flag antibody M2 (Sigma, St. Louis, MO) at 0.8 μg/mL and goat anti-mouse antibodies coupled to horseradish peroxidase (Amersham Biosciences, Uppsala, Sweden) at 0.25 μg/mL. Bound antibodies coupled to horseradish peroxidase were detected by chemoluminescence using SuperSignal substrate (Pierce) and a 2103 EnVision™ Multilabel Plate Reader (Perkin Elmer, Waltham, MA, USA).

The amounts of SNAP-tagged constructs at the cell surface were quantified by fluorescence. Briefly, HEK293 cells expressing SNAP-tagged constructs were incubated at 37 °C for 1 h with 300 nM of the SNAP-Lumi4-Tb substrate, then washed three times with Tag-Lite buffer. After excitation with a laser at 337 nm, the fluorescence of the Lumi4-Tb was collected at 620 nm for 450 µs after a 50-µs delay on a PHERAstar FS (BMG Labtech, Ortenberg, Germany)[63].

**IP measurements**. IP accumulation in HEK293 cells was measured using the IP-One HTRF kit (Cisbio Bioassays) according to the manufacturer's recommendations.

**Intracellular calcium release measurements**. Twenty-four hours after transfection with plasmids encoding the indicated GABA$_B$ subunits and a chimeric protein Gqi9, HEK-293 cells were washed with HBSS buffer (20 mM Hepes, 1 mM MgSO$_4$, 3.3 mM Na$_2$CO$_3$, 1.3 mM CaCl$_2$, 0,1% BSA, 2.5 mM probenecid) and loaded with 1 µM Ca$^{2+}$-sensitive fluorescent dye Fluo-4 AM (Molecular Probes, Eugene, OR, USA) for 1 h at 37 °C. After a wash, cells were incubated with 50 µl of buffer and 50 µl of 2 × - GABA solution at various concentrations was added after 20 s of recording. Fluorescence signals (excitation 485 nm, emission 525 nm) were measured by using the fluorescence microplate reader Flexstation (Molecular Devices, Sunnyvale, CA, USA) at sampling intervals of 1.5 s for 60 s. Data were analyzed with the program Soft Max Pro (Molecular Devices, Sunnyvale, CA, USA). Dose-response curves were fitted using Prism (GraphPad software, San Diego, CA, USA).

**Molecular modelling**. The molecular model of GB1 and GB2 7TM were generated with Modeller 9.18[64] based on the crystal structure of the mGluR1 receptor (PDB code 4OR2[65]) using the loop optimization method. The sequence of all GABA$_B$ and mGlu subtypes for rat and human species were aligned with ClustalW2[66]. Then, the sequences of mGluR1, GB1 and GB2 were extracted and used to build the model. From 100 models generated, the top ten classified by DOPE score were visually inspected, and the best scored structure with suitable loops was chosen[67].

The active and inactive dimeric arrangement of the GABA$_B$ 7TMs was built by superposition to the different dimer structures of the previously reported mGlu$_2$ model[42] until the position of GB1 and GB2 was compatible with the enhanced cross-linking found in presence of the agonist molecule. The intermediate states were generated from the mGluR2 7TM intermediate models, which are in accordance with the dynamic transition expected from the inactive to the active state. The tetrameric and oligomeric forms in active and inactive states were built by translating and rotating active and inactive GABA$_B$ dimers with PyMOL software (Palo Alto, CA, USA) in a position compatible with the enhanced cross-linking between two GABA$_{B1}$ protomers found in resting state and in presence of the agonist molecule. The oligomeric active state of a GABA$_B$ 7TM in complex with the G-protein was built using as a template the crystal structure of the active β$_2$ adrenergic receptor (PDB code 3SN6[68]). The sequence alignment was based on the structural superposition of the β$_2$ adrenergic receptor and GB2. To build the model of the active dimeric arrangement of GABA$_B$ in complex with the G-protein, the G-protein atomic coordinates (PDB code 3SN6) were transferred to the active GB2 7TM subunit.

Images based on the different states modelled from inactive to active, were calculated using UCSF Chimera software[69]. Discovery studio visualizer (Accelrys Software Inc., San Diego, CA, USA) was used for protein structure visualization and PDB file editing purposes. Multiple sequence alignment visualization and analysis were performed with Jalview software[70].

**Curve fitting and data analysis**. Curve fitting was performed using nonlinear regression using GraphPad Prism 7 software. $P$-values were determining using a paired or unpaired t test with Welch's correction.

**Reporting summary**. Further information on research design is available in the Nature Research Reporting Summary linked to this article.

## Data availability

Data supporting the findings of this manuscript are available from the corresponding authors upon reasonable request. A reporting summary for this Article is available as a Supplementary Information file. The source data underlying Figs. 2c, d, 3a, b, 4a, c, 4e, f, 6c, d, 7a, 8a, c and Supplementary Figs. 8a, 8d, 10a, 11 are provided as a Source Data file.

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

## Acknowledgements

We thank the Cisbio Bioassays for their support in providing reagents. J.L. was supported by the Ministry of Science and Technology (grant number 2018YFA0507003), the National Natural Science Foundation of China (NSFC) (grant numbers 81720108031, 81872945, 31721002 and 31420103909), the Program for Introducing Talents of Discipline to the Universities of the Ministry of Education (grant number B08029), and the Mérieux Research Grants Program of the Institut Mérieux. P.R. and J.-P.P. were supported by the Centre National de la Recherche Scientifique (CNRS), the Institut National de la Santé et de la Recherche Médicale (INSERM), and by grants from the Agence Nationale de la Recherche (ANR-09-PIRI-0011), the FRM (Equipe FRM DEQ20130326522 and DEQ20170336747). L.X. was supported by a doctoral fellowship from the French Embassy in China and X.R. by a post-doctoral fellowship from the Agència de Gestió d'Ajuts Universitaris i de Recerca (AGAUR) and the Spanish Ministry of Economy, Industry and Competitiveness (SAF2015–74132-JIN) and the PO FEDER of Catalonia 2014-2020 (Ref. 001-P-000382).

## Author contributions

L.X., Q.S., H.Z., X.R., S.G., Q.H., J.-P.P., J.L. and P.R. designed experiments; L.X., Q.S., H.Z., S.G. and Q.H. performed molecular biology, cross-linking, functional assays; L.X., Q.S., H.Z., X.R., J.-P.P., J.L. and P.R. performed data analysis; X.R. performed molecular modelling; L.X., J.L., J.-P.P. and P.R. wrote the manuscript.

## Additional information

**Competing interests:** The authors declare no competing interests.

