## [Peer Review File · Nature Communications]

Reviewers' Comments:

Reviewer #1:

Remarks to the Author:

Xue et al. study the organization of GABA-B 7TM domains within heterodimers and higher order oligomers using oxidative crosslinking of cysteine mutant protomers. They find several positions within the 7TM domains that crosslink GB1 and GB2, most notably in TM5 and TM6. Receptor activation decreases crosslinking of TM5, and increases crosslinking of TM6 (at least at one position). Correspondingly, crosslinking of TM6 promotes constitutive activity of the receptor, consistent known movements of TM6 associated with activation of other GPCRs. On one hand these are interesting studies that provide a considerable amount of information regarding crosslink-accessible sites in GB1-GB2 heterodimers and also within presumed higher order oligomers. On the other hand, the results are quite consistent with a very similar study of mGluR dimers by the same group, therefore the conceptual advance offered by this study is somewhat limited. Also, it is not particularly surprising that TM6 crosslinking is promoted by receptor activation, and vice versa, as significant movement of this helix is probably a universal feature of GPCR activation.

The authors also tend to overinterpret their results in a way that fails to fully acknowledge the limitations of crosslinking studies. Such studies are quite low resolution, as they only demonstrate sites that have the capacity to be crosslinked. Because crosslinking can trap even transient and low affinity interactions, it is not possible to unambiguously conclude that crosslinking indicates a functionally meaningful interaction. More overinterpretation appears in the title, which claims that the TM6 interface "controls activation". This is not supported by the data. Although susceptibility to crosslinking and receptor activation are positively correlated, this does not imply that TM6-TM6 interactions have any effect, much less a controlling influence, on activation under normal circumstances. More broadly, it is difficult to relate changes in crosslinking patterns to function, therefore the significance of rearrangement of protomers within oligomers is only speculative.

A serious technical issue with the manuscript that also illustrates the limitations of oxidative crosslinking is the ubiquity of crosslinking of cysteine residues placed almost anywhere within the 7TM bundle. The authors claim (pg. 6) that crosslinking occurs between cysteine residues in TM5 or TM6 and that "no such crosslinking was observed when cys were introduced in TM1, 2, 4 or 7." Inexplicably, Supplementary Figure 3 shows clearly that CuP did in fact enhance crosslinking between cys residues in both TM1 and TM7. CuP may also have enhanced crosslinking at TM1.53, TM2.52 and TM4.41, although unequal loading of protein appears to have kept the authors from making this determination. It is also clear from this figure that almost all cys mutants show substantial crosslinking without CuP, presumably due to spontaneous oxidation. For example, while control samples show almost no dimers, TM1.38 samples appear almost entirely as dimers with and without CuP treatment. It is not clear why spontaneous crosslinking at this large number of sites was ignored, or how these observations can be reconciled with the authors' model.

Despite significant spontaneous and CuP-enhanced crosslinking at a large number of sites only a few sites were tested for susceptibility to agonist stimulation. The only site that shows even modest enhancement of crosslinking by agonist is TM6.56, which incidentally does not show CuP enhancement of crosslinking in the absence of agonist (Figure 3 and S3). This suggests that some of the other spontaneously crosslinked sites throughout the 7TM bundle could also be changed by agonist. It is unfortunate that agonist effects on crosslinking were shown for only one site in TM5 and one site in TM6. The significance of agonist effects on these two sites is impossible to interpret without similar data for sites throughout the 7TM bundle. Why was TM6.56 used to demonstrate agonist enhancement of crosslinking, whereas TM6.59 was used to demonstrate crosslinking enhancement of receptor activity? Agonist effects on crosslinking of many more sites should be shown, particularly but not exclusively for TM5 and TM6. Likewise, crosslinking effects on receptor activity should be shown for more cys mutants.

In instances where crosslinking results are quantified (e.g. Figure 3a and 3b) results for all three replicates should be shown with S.D., instead of a representative single example.

Figure 4a seems to be a single experiment. This needs to be replicated.

It is not clear what data justify the conclusion that the relative movement of GB heteromers is smaller than mGluR heteromers?

Reviewer #2:

Remarks to the Author:

The manuscript by Xue et al., "Rearrangement of the transmembrane domain 6 interface controls activation of a GPCR hetero-oligomer" presents an elegant analysis of the organisation of the GABAB receptor oligomer and the structural rearrangements that occur upon activation by the endogenous ligand, GABA. By employing a Cys cross-linking approach, the authors demonstrate that there are multiple dimer interfaces within the GABAB receptor oligomer that differentially influenced and rearrange upon activation. Importantly, the structural basis for higher order oligomerisation identified is distinct from that previously observed for metabotropic glutamate receptors. The insights from this study will likely inform on the fundamental basis for heteromerisation and oligomerisation of unrelated GPCRs. The methods are described in sufficient detail to enable reproduction.

For the most part, the conclusions drawn by the authors are supported by the data presented with exception of comments below.

Comments:

The authors contend that disease-causing naturally occurring mutations of GABAB receptors may have effects on oligo/dimerization. However, the residues found to be involved in dimer interfaces do not include residues that are known to be mutated in disease. To support this hypothesis, it would be worthwhile testing whether incorporating these point mutations impacts basal or agonist induced crosslinking. Further, it would be valuable to demonstrate whether these mutants, influence the minimal dimer unit, or the tetramer/oligomer.

Figure 3, blots are representative of three experiments. However, there is no error associated with quantification of dimer % in panels a and b. Can the authors quantify the GB1-GB2 dimer % of total across the individual experiments performed and report mean +/- SD/SEM to illustrate the reproducibility of these data.

Also it's not clear why only 5.42 and 6.56 were tested here? Do other cross-linking sites within TM5 and TM6 show a similar profile with respect to activation state, i.e. are similar changes (or for Tm5 residues, a lack thereof) observed for 5.40, 5.41, 5.45 as well as 6.54, 6.55?

Similarly for Figure 6, blots are representative of three experiments. However, there is no error associated with quantification of dimer % changes in panels c and d. Please quantify from individual experiments and report mean +/- sem/SD as for Fig 3.

The authors show that stabilising the resting interface between GB1 and GB2 in Fig 4f, reduces receptor activation. If the inactive interfaces within the oligomer (e.g. GB1/GB1, Fig 6) are stabilised by Cys cross-linking, is the receptor able to couple to effectors and signal? Conversely, does stabilising the active GB1/GB1 interface in the oligomers increase constitutive activity ?

Reviewer #3:

Remarks to the Author:

The current manuscript addresses the transmembrane domain interactions in GABAB receptor. The

authors use cysteine crosslinking, fluorescent-labeled blotting and molecular modeling to identify the transmembrane interfaces of GABAB receptor heterodimer and higher-order oligomers. Given that no structural information is available for the transmembrane domain of this important G protein-coupled receptor, the current study has the potential to bring new insight for understanding the conformational rearrangement induced by receptor activation and allosteric interactions between GPCR oligomers. The manuscript can be published in Nature Communications after the authors have addressed the following issues.

1. The heterodimer formed by control GB1 and GB2 is not disulfide linked, and is expected to dissociate into single subunits on SDS PAGE. In Supplementary Figure 1a, however, faint GB1-GB2 heterodimer bands can clearly be seen on gels.

In Figure 6c and 6d, Cys mutated GB1 and non-mutated GB2 were used to identify oligomer interfaces without GB1-GB2 crosslinking. Nevertheless, GB1-GB2 heterodimer bands were observed on SDS PAGE in 4 out of 5 pairs of GB1 and GB2 (GB14.52/GB2Ctr, GB16.56/GB2Ctr, GB11.45/GB2Ctr, and GB15.42/GB2Ctr).

Could the authors provide an explanation for these results? It would also be helpful if the authors could show the full scale of each gel so that potential oligomer bands can be seen as in Figure 7.

2. In Supplementary Figure 3, GB1-GB2 dimers were detected for a majority of the symmetric Cys mutations in TM1, 2, 4, 5, 6 and 7, suggesting that GB1-GB2 dimer interactions are nonspecific.

The authors identified the residues at the GB1-GB2 dimer interface based on the increased intensity of dimer bands after treatment with CuP. However, the receptors are expressed at the cell surface in oxidative environment. If the pairs of Cys residue are located adjacent to each other at the dimer interface, disulfide-linked dimers would form with or without CuP treatment.

3. To prove that the GB1-GB2 and GB1-GB1 dimers seen on non-reducing gels are disulfide bonded, SDS PAGE of the same samples under reducing conditions need to be presented to show the dimer bands disappearing in the presence of reducing reagents.

4. Disease-linked mutations in TM6 of GB2 are cited to support the claim that TM6 is at the interface of active state. According to the model presented in Figure 1, some of the mutated residues point to the interior of the transmembrane domain, and cannot be involved in dimer interaction.

Yoo et al. found that a mutation in TM3 (A567T) is also associated with Rhett syndrome by affecting the ionic lock within the transmembrane domain. By analogy, the disease effect of at least some of the TM6 mutations could be due to similar mechanisms. For completeness, the authors need to mention the TM3 mutation in addition to the TM6 mutations in the introduction and explain the various possible mechanisms of action.

5. The results concerning the heterodimer interface in the basal state are inconsistent with the model. The authors concluded based on crosslinking data that TM5s are close in the basal state (cross-linking results in Fig 3), and so are TM4 and TM6 (Fig. 4). In the molecular model presented in Fig. 5, TM5s are at the center of the resting dimer interface, but TM4 and TM6 appear too far apart to be crosslinked by a disulfide bond.

6. The authors concluded that TM5s are less close in the active state because GB1-GB2 crosslinking between the two TM5s was largely decreased but GB1-GB1 dimers cross-linked through their TM5s increased in the presence of agonist (Figure 3a). To claim that TM4 and TM6 are close in the resting interface, the authors need to show similar change in the intensity of cross-linked dimer bands for the GB16.59/GB24.52 pair (Figure 4).

According to the schematic presented in Figure 4d, the reversed pair GB14.52/GB26.59 would be

expected to show similar intensity changes upon agonist stimulation, but no such change was seen in Supplementary Figure 7.

Functional data should be provided for GB1TM5 and GB2TM5 mutants.

7. The authors showed that CuP treatment resulted in GB1-GB1 cross-linked dimer through TM4 and TM6 in the absence of agonist (Figure 6e), and through TM1, TM5 and TM7 in the presence of agonist (Figure 6d). These symmetric interfaces cannot form based on the model proposed in Fig. 6e and 6f because the helices involved are too far apart.

8. The cross-linking results for oligomer interfaces are not consistent with the molecular model. The authors showed an oligomer band for the GB1 double Cys mutants 1.45/4.52, 4.52/6.55, 4.52/7.34, and 1.45/5.42 in Figure 7a, and 4.52/6.59 in Supplementary Figure 7. However, in the oligomer models proposed in Figure 6 and 7, the helices in none of the TM1/TM4, TM4/TM6, TM4/TM7, and TM1/TM5 pairs are close enough to be cross-linked. In contrast, the mutants that do not show oligomer bands involve helices that are closer in distance in the model. The attached diagram illustrates this point.

Additionally, the cross-linking pattern for GB1 mutant 4.52/7.34 and GB2C_{tr} are different in Figure 7 and Supplementary Figure 7, with only one showing dimer and oligomer band.

9. The title of the paper needs to be changed because a major emphasis of the paper is on the identification of oligomer interfaces of GABAB receptor that are not controlled by transmembrane domain 6 alone, and there is insufficient evidence for the involvement of this transmembrane helix in the dimer interaction in the basal state.

10. Molecular modeling is useful in understanding the cross-linking data. However, the videos need to be deleted because it seems too far stretched with the current data, especially given the many inconsistencies between the models and crosslinking data.

C

Distinct GB1-GB1 interfaces in oligomer

POINT BY POINT REPLY TO THE REFEREES

Referee #1 (Remarks to the Author):

General comments:

*Xue et al. study the organization of GABA-B 7TM domains within heterodimers and higher order oligomers using oxidative crosslinking of cysteine mutant protomers. They find several positions within the 7TM domains that crosslink GB1 and GB2, most notably in TM5 and TM6. Receptor activation decreases crosslinking of TM5, and increases crosslinking of TM6 (at least at one position). Correspondingly, crosslinking of TM6 promotes constitutive activity of the receptor, consistent known movements of TM6 associated with activation of other GPCRs. On one hand these are **interesting studies that provide a considerable amount of information regarding crosslink-accessible sites in GB1-GB2 heterodimers and also within presumed higher order oligomers.***

Specific comments :

1. *On the other hand, the results are quite consistent with **a very similar study of mGluR dimers by the same group, therefore the conceptual advance offered by this study is somewhat limited.***

We have recently reported the use of a similar approach to analyze the dimerization interface of the mGlu2 receptor that belongs to the class C GPCRs as the GABA_B receptor. This study reported TM6 as being part of the dimer interface the activated form of this homodimer. However, our present study is of strong importance for three main reasons:

- **First**, although part of the class C of the GPCR family, the **GABA_B receptor is structurally different from the other class C receptors** as it lacks the cysteine-rich domain that interconnects the VFT binding domain, to the 7TM domain. As such, the conformational changes observed in the VFT dimer (Geng et al (2013) Nature, Lecat-Guillet (2017) Cell Chem Biol) are much smaller than those reported for mGlu receptors (Pin & Bettler (2016) Nature). Because the movement reported for mGluR 7TM dimer activation is large, such movements is not expected with the GABA_B receptor raising the question whether the same dimerization interfaces are being used in this receptor. Moreover, both GABA_B subunits are linked by a coiled-coil interaction of their C-terminal intracellular domain, downstream of TM7 that may well bring some constraints influencing the final dimer interface.

- **Second, and most importantly**, whereas the mGlu receptors are strict dimers in their basal state (Maurel et al (2008) Nat Methods, Levitz et al (2016) Neuron, Moller et al (2018) Sci Rep), **GABA_B receptors are organized into oligomers of heterodimers** (Maurel et al (2008) Nat Methods, Comps-Agrar et al (2011) EMBO J, Schwenk et al (2010) Nature, Calibero et al (2013) PNAS). Then the GABA_B receptor has a far more complex quaternary structure than mGluRs. Such large complexes obviously involve multiple interfaces. It was therefore not possible to apply what was observed with the mGlu2 homodimer to the GABA_B heterocomplex. Moreover, even if conserved, do the interfaces observed in the mGlu2 homodimer are those of the GABA_{B1}-GABA_{B1} dimer, or those of the GABA_{B1}-GABA_{B2} heterodimer? This second point is **especially timely as more and more publications revealed GPCRs likely form tetramers** in the plasma membrane. Stable or not, such structures have multiple interfaces, and identifying these is essential for our understanding of the allosteric properties connecting GPCRs in such tetrameric complexes (see also our comments below to the remark #2 of this referee)

- **Last but not least**, many **mutations responsible for human diseases** have been identified in

the GABA_{B2} encoding gene (Yoo et al. (2017) *Ann Neurol*; Hamdan et al. (2017) *Am J Hum Gen*, Vuillaume et al. (2018) *Ann Neurol*), especially within the 7TM domain coding region. Surprisingly, some of these mutations were found to affect residues facing outside the 7TM bundle, raising questions on how such mutations could affect GABA_B function leading to specific brain diseases. It was then of interest to identify the possible role of these mutations in the dynamics of the heterocomplex associated with receptor activation.

2. Also, it is not particularly surprising that TM6 crosslinking is promoted by receptor activation, and vice versa, as significant movement of this helix is probably a universal feature of GPCR activation.

We agree with the referee that the movement of TM6 is probably universal for the activation of a GPCR protomer including in that of class C including the GABA_B receptor, even through its amplitude may change between GPCRs (Koehl et al. (2018) *Nature* 558:547-552). However, the relative position and the conformation of the TM6 within a dimer or oligomer remain largely unknown. Indeed, most recent models of GPCR dimers highlight either TM4-5 or TM1-H8 as the dimer interfaces, and not TM6. However, we fully agree that this TM6 is probably key to explain the cooperativity between 7TM domains in GPCR complexes. Since class C are constitutive dimers, they represent good models to investigate the possible role of TM6, not only in allowing the activation of G proteins, but also to talk to associated proteins through allosteric interaction. Because the GABA_B receptors assemble into large entities, this receptor represents a good model to identify the interplay of the interfaces involved in such GPCR complexes. We then think that our data showing that the GB1 subunits contact each other through TM6 in the inactive state, while TM6 constitutes the dimer interface of the active GB1-GB2 complex is then of major relevance to the field as it helps understand how allosteric interaction can be transmitted between subunits in a GPCR oligomer.

3. The authors also tend to overinterpret their results in a way that fails to fully acknowledge the limitations of crosslinking studies. Such studies are quite low resolution, as they only demonstrate sites that have the capacity to be crosslinked. Because crosslinking can trap even transient and low affinity interactions, it is not possible to unambiguously conclude that crosslinking indicates a functionally meaningful interaction. More overinterpretation appears in the title, which claims that the TM6 interface "controls activation". This is not supported by the data. Although susceptibility to crosslinking and receptor activation are positively correlated, this does not imply that TM6-TM6 interactions have any effect, much less a controlling influence, on activation under normal circumstances. More broadly, it is difficult to relate changes in crosslinking patterns to function, therefore the significance of rearrangement of protomers within oligomers is only speculative.

The referee is right to point out the possible limitations of the disulfide trapping strategy used in our study. However, we still think this approach gives a **rather good resolution** of the possible proximity between two residues in protein-protein interactions. A Cys-Cys disulfide bridge requires a distance below 8 Å between both cysteines. Accordingly, it was extensively used in the analysis of GPCRs such as ligand binding, dimerization and association with partner proteins. It was also used to validate crystal structures of GPCR complexes including GPCR complexed with either G-protein (Hu et al. (2010) *Nat Chem Biol* 6:541-8) or arrestin (Zhou et al. (2007) *Cell* 170:457-469). Finally, it was used to validate conformational changes in a number of ligand gated channels (Prevost et al (2012) *Nat Struct Mol Biol* 19, 642-649). However, we agree, as detailed below, that such analysis must be carried out with caution and cross-linking data must be analyzed taking into account the functional

properties of the protein.

As pointed out by the referee, the main drawback of this technique is the trapping of interactions that can be transient (but at least it shows that they can exist), some of them not being functionally relevant. Also, due to the constant conformational dynamics of the proteins and their movement in their biological environment, many interaction points between proteins can occur and can be trapped by Cys cross-linking. One cannot also exclude that cross-linking occurs during the sample preparation then after protein solubilization and unfolding before electrophoretic migration in acrylamide gels. This is why a large number of Cys-Cys disulfide trapping positions within the proteins need to be performed to see which one are the most relevant and to identify positions where no trapping can be observed (negative control). In order to relate these interactions with functional properties of the proteins, this biochemical approach needs to be coupled with functional studies. As a first analysis, we analyzed the change in cross-linking efficacy resulting from the activation of the receptor, showing that some cross-linking are favored, while others are largely decreased. This is the systematic approach that makes the conclusions stronger. Moreover, we think that the correlation between disulfide trapping and constitutive activation of the receptor (see TM6 mutant 6.56 leading to constitutive activity of the receptor) is very powerful to conclude that a contact between the top of the TM6 between GABA_{B1} and GABA_{B2} is very likely occurring in the active state of the receptor heterodimer. This is this functional consequence of the cross-linking (especially such an important gain of function) that we think is essential in our manuscript to highlight the TM6-TM6 interface as a key element in the GABA_B oligomer activation, as this interface oscillates between GB1-GB1 in the inactive state to GB1-GB2 in the active state. This is the reason why in the first version of our title we highlighted the role of the TM6 interface in "controlling" receptor activation. We agree that the term "control" is too strong and we have replaced it by "associated with" in the revised version. Altogether, we think our proposed model for GABA_B 7TM rearrangement during activation is well supported by the ensemble of cross-linking analysis (between GABA_{B1} and GABA_{B2}, as well as between GABA_{B1} and GABA_{B1}), their changes resulting from receptor activation, and the generation of a constitutively active receptor through TM6-TM6 cross-linking.

In the revised version of the manuscript (Results section, page 6 and 7), we have added few sentences to aware the readers of the limitations of the Cys cross-linking approach. Also, we have modified the title of the manuscript: "**Rearrangement of the transmembrane domain 6 interface associated with the activation of a GPCR hetero-oligomer**".

These negative comments of the referee could also come from his/her 4th remarks below. In the revised version, we bring new data to clarify these issues and try to convince that our conclusions are valid. We have re-done a large number of experiments, and repeated several times some cross-linking experiments shown in the new Figure 2 and Supp Fig 3. We included error bars and performed statistical analyses. We have change our revised manuscript accordingly (see the comments below).

4. *A serious technical issue with the manuscript that also illustrates the limitations of oxidative crosslinking is the **ubiquity of crosslinking of cysteine residues placed almost anywhere within the 7TM bundle**. The authors claim (pg. 6) that crosslinking occurs between cysteine residues in TM5 or TM6 and that "no such crosslinking was observed when cys were introduced in TM1, 2, 4 or 7." Inexplicably, Supplementary Figure 3 shows clearly that CuP did in fact enhance crosslinking between cys residues in both TM1 and TM7. CuP may also have enhanced crosslinking at TM1.53, TM2.52 and TM4.41, although unequal loading of protein appears to have*

*kept the authors from making this determination. It is also clear from this figure that almost all cys mutants show **substantial crosslinking without CuP**, presumably due to spontaneous oxidation. For example, while control samples show almost no dimers, TM1.38 samples appear almost entirely as dimers with and without CuP treatment. It is not clear why spontaneous crosslinking at this large number of sites was ignored, or how these observations can be reconciled with the authors' model.*

We thank the referee for raising these points that we tried to clarify in the revised version.

First, we would like to clarify the meaning of the GB1-GB2 dimer band (above 250 kDa). **Without CuP, this band is most probably made of heterodimers that are not covalently linked through a disulfide bridge between the GB1 and GB2 subunits.** Although we cannot exclude that part of this band can come from a spontaneous oxidation between the two subunits as suggested by the referee, when tested, most of this band is not sensitive to reducing agents such as DTT (see new results in Supp Fig. 3b, Supp Fig. 6b and Supp Fig. 8). We think it is likely due to the strong interactions between both subunits mediated by the ECD and the coiled-coil domain in the C-terminal region, in addition to the 7TMs. Indeed, this band was also observed for mGluR2 lacking the inter-subunit disulfide bond even in the presence of DTT (Xue et al (2015) Nat Chem Biol). When the C-term of mGlu2 was replaced by the C-term of the GABA_B subunits, this mGlu2 dimer band was even further increased, consistent with the GB1-GB2 C-term coiled-coil interaction stabilizing the interaction between the two mGlu2 subunits of this chimeric receptor. This is perfectly consistent with previous data by Calver et al (J Neurosci 2001) showing the importance of the coiled-coil domain in the stability of the GB1-GB2 complex. However, as well illustrated by the data presented from various experiments, the amount of GB1-GB2 dimers relative to the monomer species is variable, and we did not identify the reason for this, despite major efforts to solve this issue. This variability largely complicated the analysis of our data as well pointed out by the referee. This is why we concentrated our analysis on the CuP-induced increase in dimer formation, although we realize this is questionable, but this remains the best way we found to analyze our data.

In the revised version, **to better analyze the effect of CuP treatment on the different mutants**, we have re-done a large number of experiments, and repeated several times some cross-linking experiments shown in Figure 2 and Supp Fig 3. For all the mutants, we have quantified the % of GB1-GB2 cross-linked relative to the total quantity of GB1 subunit detected on blots. We included error bars and performed statistical analyses. These data are now shown in the new Figure 2d. Upon CuP treatment no or low increase is observed for the control GABA_B or for most Cys mutations in the TMs, except for the TM5 and TM6 mutants. Accordingly, for the control GABA_B, we think that some blots chosen in Figure 2c and Supp Fig 3 were not enough representative of the results obtained with this construct. Indeed, in general the GB1-GB2 dimer band is stronger, as observed in Supp Fig 1a. We have then replaced the control GABA_B blots by a more representative blot in Figure 2c and Supp Fig 3a. Similarly, the blots used for 1.45 and 7.34 are not the most representative ones, then we replaced them by more representative one in the revised version (no significant difference between no CuP and + CuP, see new Fig. 2d).

We would also like to clarify an important point. We do not imagine these oligomers are stable at all. Indeed, if this was the case, this would add energy to be able to escape one state and reach the other, and we do not think GABA will provide enough energy for this. Instead, we think the receptor complex is constantly oscillating from one state to another, and we expect that this is only when both an agonist and a nucleotide-free G protein are bound to the complex that it is stable. Accordingly, multiple cross-linking can occur during this process that are not necessary all compatible with a static

model. This is the reason why **we concentrated our effort in identifying positions where the cross-linking efficacy (probability) change between the active and inactive state**. These are the positions we considered as the most important ones, and are indeed important to reach our proposed model. We really think such cross-linking changes as more informative than cross-linking *per se*.

5. *Despite significant spontaneous and CuP-enhanced crosslinking at a large number of sites **only a few sites were tested for susceptibility to agonist stimulation**. The only site that shows even modest enhancement of crosslinking by agonist is TM6.56, which incidentally does not show CuP enhancement of crosslinking in the absence of agonist (Figure 3 and S3). This suggests that some of the other spontaneously crosslinked sites throughout the 7TM bundle could also be changed by agonist. It is unfortunate that agonist effects on crosslinking were shown for only one site in TM5 and one site in TM6. The significance of agonist effects on these two sites is impossible to interpret without similar data for sites throughout the 7TM bundle. Why was TM6.56 used to demonstrate agonist enhancement of crosslinking, whereas TM6.59 was used to demonstrate crosslinking enhancement of receptor activity? Agonist effects on crosslinking of many more sites should be shown, particularly but not exclusively for TM5 and TM6. Likewise, crosslinking effects on receptor activity should be shown for more cys mutants.*

The referee raised an important point. **We have quantified the agonist effects on cross-linking to all the sites of the 7TM bundle where Cys were introduced, including in TM5 and TM6**. These new data have been added in a new Figure 3c and in Supp Figure 6a. Of interest, our data show that for four positions of TM6 (6.54, 6.55, 6.56 and 6.58), the agonist increased the GB1-GB2 cross-linking consistent with GB1 and GB2 TM6s being at the heterodimer interface in the active state of the receptor (Figure 3d). In TM5, we identified two positions (5.42 and 5.45) where the agonist significantly changed (decreased) the GB1-GB2 cross-linking according to TM5s of GB1 and GB2 pointing outside of the heterodimer interface in the active conformation. Finally, no change of GB1-GB2 cross-linking was observed for the Cys engineered in TM1, TM2, TM4 and TM7 upon agonist stimulation (Supp Figure 6a), consistent with these TMs not involved in the GB1-GB2 heterodimer interface in the active state, and being accessible for receptor-receptor interaction whether the heterocomplex is in an active or inactive state.

It is difficult to know why the mutant TM6.59 but not TM6.56 shows a constitutive activity of the receptor. First, the mutant 6.56 does not strongly cross-link upon CuP treatment compared to 6.59 (see new Figure 2d). But most important, it suggests that not all interaction between TM6s in the heterodimers, but only specific one, are able to stabilize the active state of the receptor. It indicates that the active state is stabilized by specific interactions between the two subunits that are very precise.

6. *In instances where crosslinking results are quantified (e.g. Figure 3a and 3b) results for all three replicates should be shown with S.D., instead of a representative single example.*

The referee is right. In the revised version, we show the data obtained from 5 and 4 independent experiments for Figure 3a and 3b, respectively, and the corresponding mean and standard deviation. Mean and standard deviation were also plotted for all the cross-linking results (see new Figure 2d, 3c, 6c and 6d).

7. *Figure 4a seems to be a single experiment. This needs to be replicated.*

In the revised version, the new Figure 4a is a mean of three independent experiments where mean and standard deviations were plotted.

8. *It is not clear what data justify the conclusion that the relative movement of GB heteromers is smaller than mGluR heteromers?*

Our experimental data have shown a **higher probability to cross-link TM4s in mGlu2 homodimers, than in GABA_B heterodimers**. In our study, we did not obtain any specific cross-linking between GB1-TM4 and GB2-TM4 in the resting (see new Figure 2d) and active state of the receptor (Supp Figure 6a). In contrast, in the mGluR, the mGlu2 TM4s cross-link easily as observed in our previous study (Xue et al. (2015) Nat Chem Biol). Our conclusion is consistent with the observation of a smaller conformational change of the GABA_B ECD compared to mGluR ECD, as previously reported based on crystal structures (Geng et al. (2013) Nature) and FRET experiments (Lecat-Guillet et al. (2017) Cell Chem Biol).

Referee #2 (Remarks to the Author):

*The manuscript by Xue et al., "Rearrangement of the transmembrane domain 6 interface controls activation of a GPCR hetero-oligomer" presents an **elegant analysis** of the organisation of the GABAB receptor oligomer and the structural rearrangements that occur upon activation by the endogenous ligand, GABA. By employing a Cys cross-linking approach, the authors demonstrate that there are multiple dimer interfaces within the GABAB receptor oligomer that differentially influenced and rearrange upon activation. Importantly, **the structural basis for higher order oligomerisation identified is distinct from that previously observed for metabotropic glutamate receptors**. The insights from this study will likely inform on the fundamental basis for heteromerisation and oligomerisation of unrelated GPCRs. **The methods are described in sufficient detail to enable reproduction.** □ **For the most part, the conclusions drawn by the authors are supported by the data presented with exception of comments below.***

We thank the referee for his very positive comments.

Comments:□

1. *The authors contend that **disease-causing naturally occurring mutations of GABAB receptors may have effects on oligo/dimerization**. However, the residues found to be involved in dimer interfaces do not include residues that are known to be mutated in disease. To support this hypothesis, it would be worthwhile testing whether incorporating these point mutations impacts basal or agonist induced crosslinking. Further, it would be valuable to demonstrate whether these mutants, influence the minimal dimer unit, or the tetramer/oligomer.*

We agree with the referee regarding the importance of analyzing the effect of these human genetic mutations on the heterodimer unit and oligomer. Accordingly, we have introduced the **genetic mutations S694^{6,42} in our rat GB2 constructs** (equivalent to genetic mutation S695^{6,42} in human GB2), that produced a strong constitutive activity of the GABA_B receptor as recently reported (Vuillaume et al. (2018) Ann Neurol). Of interest, we found that, **in the absence of agonist, this mutation stabilizes the active interface of the heterodimer** unit mediated by both TM6s as measured by the increased GB1-GB2 cross-linked upon CuP treatment (see new Figure 8a and b). In addition, **in the basal state (absence of agonist) this mutation stabilizes the active interface between the GB1 subunits in the oligomer**, as measured by the efficient crosslinking between the

GB1 TM5s upon CuP treatment (see new Figure 8c and d). Altogether these data are consistent with a constitutive activity of the receptor induced by this mutation, associated with the stabilization of an oligomer organized in an active assembly.

2. *Figure 3, blots are representative of three experiments. However, there is no error associated with quantification of dimer % in panels a and b. Can the authors quantify the GB1-GB2 dimer % of total across the individual experiments performed and report mean +/- SD/SEM to illustrate the reproducibility of these data.* □

The referee is right. In the revised version, we show the data obtained from 5 and 4 independent experiments for Figure 3a, 3b and 3c, respectively, and the corresponding mean and standard deviation. Mean and standard deviation were also plotted for all the cross-linking results in the new Figure 3c and Supp Figure 6a.

3. *Also it's not clear why only 5.42 and 6.56 were tested here? Do other cross-linking sites within TM5 and TM6 show a similar profile with respect to activation state, i.e. are similar changes (or for Tm5 residues, a lack thereof) observed for 5.40, 5.41, 5.45 as well as 6.54, 6.55?*

The referee raised an important point. We have quantified the agonist effects on cross-linking to all the sites of the 7TM bundle where Cys were introduced, including in TM5 and TM6. These new data have been added in a new Figure 3c and in Supp Figure 6a. Of interest, our data show that for four positions of TM6 (6.54, 6.55, 6.56 and 6.58), the agonist increased the GB1-GB2 cross-linking consistent with GB1 and GB2 TM6s being at the heterodimer interface in the active state of the receptor (Figure 3d). In TM5, we identified two positions (5.42 and 5.45) where the agonist significantly changed (decreased) the GB1-GB2 cross-linking according to TM5s of GB1 and GB2 pointing outside of the heterodimer interface in the active conformation. Finally, no change of GB1-GB2 cross-linking was observed for the Cys engineered in TM1, TM2, TM4 and TM7 upon agonist stimulation (Supp Figure 6a), consistent with these TMs not involved in the GB1-GB2 heterodimer interface in the active state, and being accessible for receptor-receptor interaction whether the heterocomplex is in an active or inactive state.

4. *Similarly for Figure 6, blots are representative of three experiments. However, there is no error associated with quantification of dimer % changes in panels c and d. Please quantify from individual experiments and report mean +/- sem/SD as for Fig 3.*

The referee is right. In the revised version, mean and standard deviation were also plotted for all the cross-linking results in Figure 6c and 6d.

5. *The authors show that stabilising the resting interface between GB1 and GB2 in Fig 4f, reduces receptor activation. If the inactive interfaces within the oligomer (e.g. GB1/GB1, Fig 6) are stabilised by Cys cross-linking, is the receptor able to couple to effectors and signal? Conversely, does stabilising the active GB1/GB1 interface in the oligomers increase constitutive activity ?*

This is an interesting point. We have tried to stabilize either the inactive or active state of the oligomer by stabilizing the GB1-GB1 interface with Cys. We have introduced either single or double cysteine mutations in GB1. For example, we have performed the cross-linking of GB1^{6.59} and GB2^{Ctrl}, and GB1^{4.52/6.59} and GB2^{Ctrl}, to stabilize the inactive state of the receptor. Unfortunately, we could not observe any functional effect of these mutants, neither a decrease nor an increase in coupling

efficacy. This could be due to the low efficiency of the GB1-GB1 cross-linking, at least for the inactive state of the oligomer. Indeed, to get a significant effect, a large proportion of the subunits must be cross-linked, making it quite difficult to achieve. In our previous study, to be able to see a decrease in the coupling efficacy of mGlu2 homodimers stabilized in their inactive conformation, we had to introduce two Cys, one in TM4 and the other in TM5 to ensure that a large proportion of the dimers were cross-linked at that interface (Xue et al. (2015) Nat Chem Biol). Only under these conditions we observe a large decrease in the coupling capacity of this receptor.

Referee #3 (Remarks to the Author):

The current manuscript addresses the transmembrane domain interactions in GABAB receptor. The authors use cysteine crosslinking, fluorescent-labeled blotting and molecular modeling to identify the transmembrane interfaces of GABAB receptor heterodimer and higher-order oligomers. Given that no structural information is available for the transmembrane domain of this important G protein-coupled receptor, the current study has the potential to bring new insight for understanding the conformational rearrangement induced by receptor activation and allosteric interactions between GPCR oligomers. The manuscript can be published in Nature Communications after the authors have addressed the following issues.

We thank the referee for her/his positive comments.

Comments

1. The heterodimer formed by control GB1 and GB2 is not disulfide linked, and is expected to dissociate into single subunits on SDS PAGE. In Supplementary Figure 1a, however, faint GB1-GB2 heterodimer bands can clearly be seen on gels.

In Figure 6c and 6d, Cys mutated GB1 and non-mutated GB2 were used to identify oligomer interfaces without GB1-GB2 crosslinking. Nevertheless, GB1-GB2 heterodimer bands were observed on SDS PAGE in 4 out of 5 pairs of GB1 and GB2 (GB14.52/GB2Ctr, GB16.56/GB2Ctr, GB11.45/GB2Ctr, and GB15.42/GB2Ctr).

Could the authors provide an explanation for these results? It would also be helpful if the authors could show the full scale of each gel so that potential oligomer bands can be seen as in Figure 7.

We thank the referee for raising this point that was not explained in the manuscript. It is now clearly stated in the revised version.

The referee is right, in the absence of CuP, no disulfide bridge should exist between GB1 and GB2 subunit, and they should be observed as single and monomeric subunits on SDS-PAGE. Even if we cannot exclude a spontaneous oxidation between the two subunits. As indicated in our answer to referee's #1 comments, we think the **GB1-GB2 dimer band is mostly made of heterodimer that are not covalently linked through a disulfide bridge between the GB1 and GB2 subunits**. Indeed, most of this band is not sensitive to reducing agents such as DTT (see new results in Supp Fig. 3b, Supp Fig. 6b and Supp Fig. 8). It is probably difficult to separate the two subunits of the GABA_B heterodimer entity in the whole population of receptors even in the absence of non-covalent association between GB1 and GB2 subunits. This is mainly, but not exclusively, due to the presence of a coiled-coil domain at the level of their C terminal domains (Calver et al. (2001) J Neurosci). Such dimer band without disulfide linked subunits were also observed for other class C GPCRs such as mGluRs. Indeed, this band was observed for mGluR2 lacking the inter-subunit disulfide bond even in the presence of DTT (Xue et al (2015) Nat Chem Biol). And when the C-terminal regions of mGluR2

were replaced by the C-term of GABA_B containing the coiled-coil domain, this mGlu2 dimer band was even further increased, further indicating that the coiled-coil domain stabilizes the interactions between the two mGlu2 subunits of this chimeric receptor.

Since this GB1-GB2 dimer band is observed in most of the experiments (see for example Figure 2c), to avoid the readers to be confused, we have replaced the blot GB1-7.34/GB2-Ctr (Figure 6d) by one where GB1-GB2 dimer band is present.

To get rid of the GB1-GB2 dimer band in absence of CuP, it would probably require to optimize experimental conditions and buffer using specific combinations of detergents. This may prove to be difficult because of the high affinity coiled-coil interaction between the GB1 and GB2 subunits (Burmakina et al., PNAS, 2014). But unfortunately, we have not succeeded in optimizing this step.

The SDS-PAGE gels shown in Figure 6c and 6d were already full size. But unfortunately, they were run in conditions where the potential oligomer bands cannot be seen (6 % polyacrylamide, with a ratio 29:1 acryl:bisacryl-amide), in contrast to gels shown in Figure 7 (6 % polyacrylamide, with a ratio 59:1 acryl:bisacryl-amide).

2. In Supplementary Figure 3, GB1-GB2 dimers were detected for a majority of the symmetric Cys mutations in TM1, 2, 4, 5, 6 and 7, suggesting that GB1-GB2 dimer interactions are nonspecific.

The authors identified the residues at the GB1-GB2 dimer interface based on the increased intensity of dimer bands after treatment with CuP. However, the receptors are expressed at the cell surface in oxidative environment. If the pairs of Cys residue are located adjacent to each other at the dimer interface, disulfide-linked dimers would form with or without CuP treatment.

A similar issue was raised by the Referee #1 (see Point 4). As indicated above, GB1 and GB2 strongly interact through their coiled-coil domain, such that a GB1-GB2 dimer band can easily be observed independently of any disulfide bond. However, for the cysteine engineered GB1-GB2 dimer band, the higher signal observed after CuP treatment is indicative that a disulfide bond can be formed. As shown in the new Fig 2b only few Cys mutants lead to an increase intensity of the dimer bands after treatment with CuP, associated with a decrease in the relative amount of the GB1 monomer band (Supp Figure 3a).

We agree with the referee that the extracellular region is an oxidative environment, but the Cys were introduced in the transmembrane domain of the receptor and in this local environment the oxidation of the Cys residues is not efficient and usually required the addition of an oxidative reagent such as CuP, as shown here and in many other studies for GPCRs and other transmembrane proteins (See Javitch's articles and others such as Guo et al. (2008) EMBO J, for example).

3. To prove that the GB1-GB2 and GB1-GB1 dimers seen on non-reducing gels are disulfide bonded, SDS PAGE of the same samples under reducing conditions need to be presented to show the dimer bands disappearing in the presence of reducing reagents.

The referee is right. We used the reducing agent dithiothreitol (DTT) to show that the CuP-induced cysteine cross-linkings are reversible (see new results in Supp Fig. 3b, Supp Fig. 6b and Supp Fig. 8). We observed that the CuP treated GB1-GB2 and GB1-GB1 bands are sensitive to DTT. However, the GB1-GB2 band does not totally disappear even in the absence of CuP, as this heterodimer is known

to be stabilized by the coiled-coil domain located in their C terminal part (Calver et al. (2001)), as discussed above. The GB1-GB1 dimer band also does not totally disappear after DTT treatment, and it might be due to none covalent but strong interactions remain between GB1 subunits (Vuilleumure et al (2005) Biochem J).

4. Disease-linked mutations in TM6 of GB2 are cited to support the claim that TM6 is at the interface of active state. According to the model presented in Figure 1, some of the mutated residues point to the interior of the transmembrane domain, and cannot be involved in dimer interaction.

Yoo et al. found that a mutation in TM3 (A567T) is also associated with Rhett syndrome by affecting the ionic lock within the transmembrane domain. By analogy, the disease effect of at least some of the TM6 mutations could be due to similar mechanisms. For completeness, the authors need to mention the TM3 mutation in addition to the TM6 mutations in the introduction and explain the various possible mechanisms of action.

This is an interesting point raised by the referee. It is true that in the first version of our manuscript we concentrated our presentation on the mutations affecting side chains oriented outward the 7TM bundle, but it is also true that other mutations are oriented inward and affect the conformation of the GB2 7TM leading to constitutive activity. These were perfectly identified in previous studies and it was possible to provide a structural explanation for their effects (Yoo et al. (2017) Ann Neurol; Vuillaume et al. (2018) Ann neurol). However, the reason for the consequences of those oriented outside the 7TM bundle were far from clear. We think our study provides a reasonable explanation for these. To further strengthen this point, we have investigated how the genetic mutations in TM6 could influence the interface in the heterodimer unit and oligomer in the basal state since these mutations were shown to induce a strong constitutive activity of the receptor (Vuillaume et al. (2018) Ann Neurology). The mutation GB2 S694I^{6,42} (equivalent to the genetic mutation S695I^{6,42} in human GB2) that produced a strong constitutive activity receptor was used. In the absence of agonist, this mutation stabilized the active interface of the heterodimer unit mediated by both TM6s as measured by the increase GB1-GB2 cross-linked upon CuP treatment (see new Figure 8a and b). In addition, this mutation stabilizes the active interface between the GB1 subunits in the oligomer in the basal state, as measured by a strong crosslinking between the GB1 TM5s upon CuP treatment (see new Figure 8c and d). Altogether these data are consistent with a constitutive activity of the receptor induced by this mutation.

5. The results concerning the heterodimer interface in the basal state are inconsistent with the model. The authors concluded based on crosslinking data that TM5s are close in the basal state (cross-linking results in Fig 3), and so are TM4 and TM6 (Fig. 4). In the molecular model presented in Fig. 5, TM5s are at the center of the resting dimer interface, but TM4 and TM6 appear too far apart to be crosslinked by a disulfide bond.

The heterodimer interface has to be seen as dynamics in its basal state. Even if TM5 is at the center of the resting interface, TM4, TM5 and TM6 belong to the same face of the GABA_B 7TM, then the GB1-TM4:GB2-TM6 cross-linking could be explained by the dynamics of the interface. Indeed, we do not believe the relative positions of the protomers are very stable (unless stabilized by both an agonist and a nucleotide free G-protein). If such conformations were stable, this would require more energy to go from one state to the other. It is unlikely that GABA by itself can bring enough energy for this. Instead, we see the oligomer being constantly oscillating between these two states, allowing a Cys residue to contact its symmetric Cys during such a process. This is consistent with the constant oscillation between the active and inactive state of the mGlu2 VFT dimer that we reported (Olofsson et al. (2014) Nat Commun; Vafabakhsh et al (2015) Nature 524, 497-501; Levitz et al (2016) Neuron).

This is with this idea in mind that **we considered the cross-linking events in a probabilistic manner, and this is why we concentrated our efforts in analyzing the change in probability (efficacy) of the cross-linking of a given position depending on the resting or active state.**

6. *The authors concluded that TM5s are less close in the active state because GB1-GB2 cross-linking between the two TM5s was largely decreased but GB1-GB1 dimers cross-linked through their TM5s increased in the presence of agonist (Figure 3a). To claim that TM4 and TM6 are close in the resting interface, the authors need to show similar change in the intensity of cross-linked dimer bands for the GB16.59/GB24.52 pair (Figure 4).*

According to the schematic presented in Figure 4d, the reversed pair GB14.52/GB26.59 would be expected to show similar intensity changes upon agonist stimulation, but no such change was seen in Supplementary Figure 7. □ Functional data should be provided for GB1TM5 and GB2TM5 mutants.

According to the referee's comment, we have further investigated how the interaction between GB1-TM6 and GB2-TM4 changes during receptor activation. In the new Supp Figure 7e, we show that the cross-linking between these two TMs in the basal state is slightly reduced upon agonist stimulation, in agreement with the decrease of GB1-GB2 TM5s interface during activation.

For the reversed pair GB1-4.52 / GB2-6.59, the cross-linking induced by CuP in basal state is only slightly decreased by agonist stimulation. It is shown now in the new Supp Figure 7e (the previously "Supplementary Fig.7" as misunderstood by the referee).

Finally, we provide the functional data for GB1-TM5 and GB2-TM5 mutants in the new Supp Figure 5d. These results show that cross-linking at the TM5s heterodimer interface does not impair functional activity of the receptor. We think it is because in the GB1-TM5:GB2-TM5 cross-linking experiments, the oligomer are stabilized by GB1-GB1 cross-linking through GB1-TM5 (Figure 3a), and such oligomers with GB1-GB1 TM5 interface have a conformation closer to the active state of the receptor. As a consequence, it is not possible to stabilize the basal state of the heterodimer unit through TM5s cross-linking between GB1 and GB2 subunits. The best stabilization of this basal state was so far obtained by cross-linking GB1-6.59 and GB2-4.52 (Figure 4d-f).

7. *The authors showed that CuP treatment resulted in GB1-GB1 cross-linked dimer through TM4 and TM6 in the absence of agonist (Figure 6e), and through TM1, TM5 and TM7 in the presence of agonist (Figure 6d). These symmetric interfaces cannot form based on the model proposed in Fig. 6e and 6f because the helices involved are too far apart.*

Based on the 3D model of the GABA_B oligomer in the inactive and active state (now Figure 9 in the revised version), the symmetric interface TM4-TM4 and TM6-TM6 between GB1-GB1 are closed in the inactive state. In the active state, the symmetric interface TM1-TM1, TM5-TM5 and TM7-TM7 between GB1-GB1 should be also possible, if we consider the dynamics of the receptor conformation, as indicated in our response to Point 5.

8. *The cross-linking results for oligomer interfaces are not consistent with the molecular model. The authors showed an oligomer band for the GB1 double Cys mutants 1.45/4.52, 4.52/6.55, 4.52/7.34, and 1.45/5.42 in Figure 7a, and 4.52/6.59 in Supplementary Figure 7. However, in the oligomer models proposed in Figure 6 and 7, the helices in none of the TM1/TM4, TM4/TM6, TM4/TM7, and TM1/TM5 pairs are close enough to be cross-linked. In contrast, the mutants that do not show oligomer bands involve helices that are closer in distance in the model. The attached diagram*

illustrates this point.

Additionally, the cross-linking pattern for GB1 mutant 4.52/7.34 and GB2Ctr are different in Figure 7 and Supplementary Figure 7, with only one showing dimer and oligomer band.

We think the referee has misunderstood the results regarding the oligomerization state of the receptor, probably because the Figure 7 was not clear enough. In Figure 7a, while the GB1-GB1 dimer band corresponds to two subunits of GB1 being cross-linked, the oligomer band corresponds to at least three GB1 subunits cross-linked in the same complex. In these oligomers, the three GB1 are most probably cross-linked by two different interfaces, involving TM4-TM5 on one side and TM1-TM6-TM7 on the other side. For example, when using the GB1 1.45/4.52 double mutant, we expect two GB1 subunits will be cross-linked through the 4.52 Cys, leaving the Cys 1.45 available for cross-linking an additional GB1 subunit, likely through a 1.45-1.45 bridge. We have clarified this Figure 7 by adding pictograms below panel a, to better illustrate the cross-linking of the GB1 subunits in the different conditions.

Regarding the GB1 mutant 4.52/7.34 + GB2 control, we are sorry to have made a mistake in the Supp Figure 7, where “4.52/7.34” should be replaced by “5.40/7.34”. The new figure (now Supp Fig. 9a) has been modified accordingly.

9. The title of the paper needs to be changed because a major emphasis of the paper is on the identification of oligomer interfaces of GABAB receptor that are not controlled by transmembrane domain 6 alone, and there is insufficient evidence for the involvement of this transmembrane helix in the dimer interaction in the basal state.

As also stated in reply to Referee #1 comments, we agree that we need to be more cautious in our conclusion. Although our data demonstrate that TM6 cross-linking between GB1 and GB2 leads to a constitutive activation of the receptor, we agree that the term "control" in the title is too strong and we have replaced it by “associated with” in the revised version. Accordingly, we have changed our title into: **“Rearrangement of the transmembrane domain 6 interface associated with the activation of a GPCR hetero-oligomer”**.

10. Molecular modeling is useful in understanding the cross-linking data. However, the videos need to be deleted because it seems too far stretched with the current data, especially given the many inconsistencies between the models and crosslinking data.

We understand the caution of the referee regarding these videos. However, as well illustrated by our answer to her/his Points 5 and 7, the models of the inactive and active oligomers show a very static view of this receptor complex, leading to a possible misunderstanding of our data. By giving access to these movies we tend to illustrate that this oligomer is not stable, and is likely oscillating between these two states, offering additional possibilities on the way for Cys cross-linking. Then, even though we agree that these movies represent an overstatement of our data if taken as a possible reality, they are useful to any reader to better understand our data.

Reviewers' Comments:

Reviewer #1:

Remarks to the Author:

The authors have performed several additional experiments to address many of the concerns raised by all three reviewers. The manuscript is significantly improved in several areas, starting with a more appropriate title.

One outstanding concern that has not been fully addressed is the acknowledgement of apparent crosslinking at sites other than those emphasized by the authors. Two reviewers noted that introduction of cysteines in most of the tested locations increased the proportion of GB1-GB2 dimers, even in the absence of CuP. In the revision and their rebuttal, the authors argue that these dimers represent non-covalent interactions that are insensitive to DTT and therefore are not due to disulfide bonds.

There are several concerns with this presentation of the data. First, the original Supplementary Figure 3a showed a dramatic increase in GB1-GB2 dimers for most of the samples. This is less obvious in the revised Supplementary Figure 3a because the authors have substituted different "representative" blots. In the revision and in the rebuttal the authors claim to have quantified the dimer fraction (dimer/total GB1), but they do not provide this information, and instead show only the CuP-induced percent change. This obscures background crosslinking.

1. The absolute dimer fraction needs to be provided for all of the cysteine mutants shown in Supplementary Figure 3a. The authors already have these numbers.

In the revision and rebuttal the authors claim that most of the spontaneous dimer bands resist DTT and refer to Supplementary Figure 3b. However, in this figure only the mutants that showed CuP-induced enhancement were studied, and both spontaneous and CuP-induced dimer fractions were diminished by DTT, and both only partially.

2. Several of the mutants that appear to be significantly dimeric in the absence of CuP (1.38, 4.30, 7.29) need to be tested for sensitivity to DTT.

3. The authors refer to "spontaneous oxidation" but in the very next sentence claim that this is not due to disulfide bond formation. What is being oxidized?

On Pg. 6 and several places further into the manuscript the authors note that DTT does not completely reverse the increase in GB1-GB2 dimer fraction induced by CuP. A good example of this is Supplementary Figure 3b, where site 6.55 seems to be hardly affected by DTT. The authors repeatedly suggest that the persistence of these bands is related to "none covalent" interactions between protomers, but this does not explain why such bands are only seen in these cysteine mutants and only after CuP.

4. The authors should explain how bands that appear after CuP and are attributed to oxidative crosslinking are largely insensitive to reducing agents.

On Pg. 10 the authors write that "in the presence of agonist, a strong increase in GB1-GB1 cross-linking is observed for Cys located in TM1, TM5 and TM7, but not for TM4 and TM6." This leaves the impression that no change was seen for TM4, when in fact a highly (***) significant increase was observed. Supplementary Figure 8 also appears to show an increase at TM6.

5. The description of these data needs to be revised to more accurately describe what was observed.

All of the GB1mutant-GB2control heterodimers shown in Supplementary Figure 8 appear to be sensitive to DTT, even though only one of the protomers bears an introduced reactive Cys.

6. Can the authors provide some explanation for this observation?

Taken together it seems likely that the control constructs, despite having some C>A mutations, still retain some reactive cysteine residues that spontaneously form disulfide bonds with low efficiency. This is enhanced by introduction of cysteines in many locations. This complicates the authors' analysis by overlaying "noise" on the enhanced crosslinking signals they introduce by adding targeted cysteines in specific locations, increasing the oxidative environment with CuP, and changing receptor activation state. I think the authors' job would be much easier, and the manuscript would be much improved, if this possibility was directly addressed and either convincingly ruled out or simply acknowledged as an overlying factor. Importantly, this would not invalidate many of the more important conclusions of the manuscript.

Reviewer #2:

Remarks to the Author:

The revised manuscript has satisfactorily addressed my previous comments/queries.

Minor point:

Individual data points are not shown for the bar graphs in Figures 3c and 4c as per editorial policy checklist.

Reviewer #3:

Remarks to the Author:

1. The revised title is almost the same as the original one. The authors need to remove mention of transmembrane domain 6 from the title especially if they think the dimer association process is dynamic and can involve different helices in both the inactive and active states. In addition, GB1 oligomers are a significant part of the manuscript, and the oligomeric interfaces can involve TM1, 4, 5, 6 and 7.

2. The videos need to be removed because there is not sufficient evidence supporting a three-dimensional model of the heterodimer let alone the oligomers, and the authors agree that the videos are an overstatement of their data. The videos also do not reflect the dynamics of receptor conformation.

3. Regarding the authors' explanation of persistent GB1-GB2 dimer bands, it is highly unlikely that the presence of a coiled-coil domain would have caused the appearance of these dimer bands under denaturing and reducing conditions. Although coiled-coil interactions are of high affinity, they are still non-covalent, and cannot be maintained once the proteins are denatured. The authors cite a structure paper (Burmakina et al., PNAS, 2014) to show the high affinity coil-coil interaction. However, in that same paper, the western blots of wild-type and various coiled-coil mutants of the receptor only appeared as monomeric bands, indicating that coil-coiled interactions are not sufficient to retain heterodimers once the proteins are denatured. By the same token, the interaction between the ECDs is non-covalent, and is not strong enough to produce the dimer band seen on most gels in this study.

The persistent dimer bands most likely resulted from non-specific crosslinking or non-specific association of the receptor subunits upon denaturation. I think the authors need to indicate this when they explain the limitation of the crosslinking method.

4. On page 9 of the authors' response, it was indicated that the oligomer band could only be seen when the gels were run with a ratio of 59:1 acrylamide:bisacrylamide, not with a ratio of 29:1 acrylamide:bisacrylamide. Does this mean that the oligomer and heterodimer bands co-migrated at a 29:1 acrylamide:bisacrylamide ratio? If this is the case, it would be difficult to determine whether the difference in band intensity was due to changes in the amount of cross-linked heterodimer or oligomer. The authors should consider repeating the gels with 59:1 acrylamide:bisacrylamide to separate the dimer and oligomer bands.

5. Several models for GB1 oligomers have been provided in Figure 7 to explain the different ways three GB1 molecules may interact to form an oligomer. However, the arrangements look quite different from one another, with some potentially precluding the binding of GB2 to at least one GB1 molecule due to steric interaction. These models appear artificial and also suggest that the oligomeric interface is formed in random orientations. Do the authors have any functional data that support the physiological relevance of the various GB1 oligomers?

Point by point answers to the reviewers' comments:

Reviewer #1 (Remarks to the Author):

The authors have performed several additional experiments to address many of the concerns raised by all three reviewers. The manuscript is significantly improved in several areas, starting with a more appropriate title.

We wish to thank the referee for recognizing the improvements made on the first revision of our manuscript.

One outstanding concern that has not been fully addressed is the acknowledgement of apparent crosslinking at sites other than those emphasized by the authors. Two reviewers noted that introduction of cysteines in most of the tested locations increased the proportion of GB1-GB2 dimers, even in the absence of CuP. In the revision and their rebuttal, the authors argue that these dimers represent non-covalent interactions that are insensitive to DTT and therefore are not due to disulfide bonds.

There are several concerns with this presentation of the data. First, the original Supplementary Figure 3a showed a dramatic increase in GB1-GB2 dimers for most of the samples. This is less obvious in the revised Supplementary Figure 3a because the authors have substituted different "representative" blots. In the revision and in the rebuttal the authors claim to have quantified the dimer fraction (dimer/total GB1), but they do not provide this information, and instead show only the CuP-induced percent change. This obscures background crosslinking.

1. The absolute dimer fraction needs to be provided for all of the cysteine mutants shown in Supplementary Figure 3a. The authors already have these numbers.

The referee is right when asking for this information. We apologize for not providing these data in the revised version. We now include a new bar graph in which we represent the ratio between GB1-GB2 dimer over total of GB1 subunit obtained in any individual experiment for the "control" constructs and every mutant examined (**new Supp Fig. 4**). As it can be seen, this ratio is variable for any of the constructs examined. As detailed below, we think, as also indicated by referee #3, that this is due to "non-specific crosslinking or non-specific association of the receptor subunits upon denaturation", a phenomenon that is difficult to control and that is then variable from experiment to experiment. We think this representation nicely complements the data presented in Supp Fig. 3, where only one blot of each case is presented, showing the variability of the relative amount of the dimer band. This is best illustrated with the data shown with the "control" receptor for which we had a large number of individual data as this was used as a control in every blot. As indicated in our answer to referee #3's comments, we have better explained this in the text (see page 6 and 7 of the ms), as this is clearly one of the limit of the Cys cross-linking strategy.

To make it short here, the denaturation of membrane proteins often leads to non-specific association (entangling of the protein chains due to hydrophobic interaction between them) that resist to SDS and gel migration. We think that this is made easier in the case of the GABA_B receptor due to the presence of the coiled-coil domain that increases the affinity between the two subunits before denaturation (Calver et al., J Neurosci 2001) (see our answer to point 3 of referee #3 below for more information). Such non-specific association may also be facilitated by Cys crosslinking of the proteins in the plasma membrane such that the proteins remains associated even after DTT treatment of the denatured samples.

In the revision and rebuttal the authors claim that most of the spontaneous dimer bands resist DTT and refer to Supplementary Figure 3b. However, in this figure only the mutants that showed CuP-induced enhancement were studied, and both spontaneous and CuP-induced dimer fractions were diminished by DTT, and both only partially.

2. Several of the mutants that appear to be significantly dimeric in the absence of CuP (1.38, 4.30, 7.29) need to be tested for sensitivity to DTT.

As indicated in the additional figure representing the dimer/total ratio for any individual experiments, a large variability is observed, such that no clear conclusion can be drawn from this ratio from a single experiment. The referee should now refer to the new bar graph presented in Supp Fig. 4, showing that we cannot conclude there is more dimers in the absence of CuP for mutants 1.38, 4.30 and 7.29. We however tested the DTT sensitivity for these dimer bands and detected a significant but very partial inhibition (**Rebuttal Fig. 1**), in agreement with our view that, in the absence of CuP, the dimer band likely results from a non-specific SDS resistant association of the subunits, part of these still resulting from a non-specific crosslinking (see also our answer to point 6 below).

Rebuttal Fig. 1: The intensity of the GB1-GB2 dimer band relative to the total signal was measured from five independent blots for the 1:38, 4:30 and 7:29 mutants, under control condition or after CuP treatment, with or without DTT treatment before loading the samples on the gel. One typical blot is shown below. Paired t test * $p < 0.05$, ** $p < 0.01$

3. The authors refer to “spontaneous oxidation” but in the very next sentence claim that this is not due to disulfide bond formation. What is being oxidized?

In our rebuttal, we mentioned that part of the GB1-GB2 dimers can originate from spontaneous oxidation, and then mention that “most of this band is not sensitive to reducing agents”. As indicated above, we think that dimers assembled in the plasma membrane are more prone to remain associated after denaturation, such that they remain associated in the acrylamide gel even if the disulfide bridges are reduced. Denaturation of folded proteins is also prone to expose Cys residues that can be oxidized during the sample preparation. We think such crosslinking of already denature proteins is less prone to non-specific association, and then is more prone to dissociation after DTT treatment. The text has been clarified to indicate that non-specific SDS resistant association and/or crosslinking can happen between already associated, but not covalently linked membrane proteins during denaturation (page 6 and 7).

On Pg. 6 and several places further into the manuscript the authors note that DTT does not completely reverse the increase in GB1-GB2 dimer fraction induced by CuP. A good example of this is Supplementary Figure 3b, where site 6.55 seems to be hardly affected by DTT. The authors repeatedly suggest that the persistence of these bands is related to “none covalent” interactions between protomers, but this does not explain why such bands are only seen in these cysteine mutants and only after CuP.

4. The authors should explain how bands that appear after CuP and are attributed to oxidative crosslinking are largely insensitive to reducing agents.

DTT treatment is performed after denaturation of the membrane proteins for 10 min, just before loading the gel. We think any crosslinking between the two subunits before denaturation may favor non-specific association between the proteins even after the 10 min

DTT treatment performed before loading. As a consequence, we think the two entangled protein chains remain associated even though they are no longer crosslinked.

On Pg. 10 the authors write that “in the presence of agonist, a strong increase in GB1-GB1 cross-linking is observed for Cys located in TM1, TM5 and TM7, but not for TM4 and TM6.” This leaves the impression that no change was seen for TM4, when in fact a highly (****) significant increase was observed. Supplementary Figure 8 also appears to show an increase at TM6.

5. The description of these data needs to be revised to more accurately describe what was observed.

The referee is perfectly right, and we must apologize for this non-accurate description of our data. We have now carefully check our manuscript to make sure that our results section matches the data obtained and illustrated in both the main and sup Figures.

All of the GB1mutant-GB2control heterodimers shown in Supplementary Figure 8 appear to be sensitive to DTT, even though only one of the protomers bears an introduced reactive Cys.

6. Can the authors provide some explanation for this observation?

The referee is perfectly right, the intensity of the GB1-GB2 dimer band is often decreased after DTT treatment, as can be seen in Supp Fig. 8. This is also the case with the “Control” combination (**Rebuttal Fig. 2**). We think this is due to non-specific crosslinking occurring during or after receptor denaturation due to the exposure of native Cys residues. Such crosslinking occurring after denaturation is likely less prone to favor non-specific protein association, and then less prone to remain associated after DTT-treatment. In agreement with this hypothesis, the intensity of this band is not increased upon CuP treatment that is being performed on intact cells, the CuP oxidation being stopped using NEM before starting the denaturation process. It is possible that the non-specifically crosslinked subunits formed after denaturation are more sensitive to DTT treatment. Indeed, the subunits crosslinked in the cell membrane are more likely to remain associated after denaturation such that DTT treatment at that step does not lead to their efficient dissociation. As indicated above, this limitation of the Cys crosslinking approach is now indicated in our manuscript.

Rebuttal Fig. 2: Blots illustrating GB1 monomer and GB1-GB2 dimers of control subunits under the indicated condition. Note that DTT treatment is being performed with 10 mM DTT for 10 min after denaturation, just before loading the samples on the gel. Data are from three separate experiments in which the dimer over monomer ratios are different.

Taken together it seems likely that the control constructs, despite having some C>A mutations, still retain some reactive cysteine residues that spontaneously form disulfide bonds with low efficiency. This is enhanced by introduction of cysteines in many locations. This complicates the authors’ analysis by overlaying “noise” on the enhanced crosslinking signals they introduce by adding targeted cysteines in specific locations, increasing the oxidative environment with CuP, and changing receptor activation state. I think the authors’ job would be much easier, and the manuscript would be much improved, if this possibility was directly addressed and either convincingly ruled out or simply acknowledged as an overlying factor. Importantly, this would not invalidate many of the more important conclusions of the manuscript.

We thank the referee for pointing that our data support many of our important conclusions. As requested by the referee, we now acknowledge that non-specific, SDS-resistant dimer bands is a clear drawback of our approach (see page 6 and 7 in the ms). We still think the

CuP-induced dimer and oligomer bands, and especially the changes observed after agonist treatment is really informative on the interface changes occurring during the activation of the GABA_B receptor complex. As this parameter has been mainly used to reach our conclusions, the referee is right when indicating that our data largely support our model of the relative movement between the GABA_B receptor 7TMs occurring upon receptor activation.

Reviewer #2 (Remarks to the Author):

The revised manuscript has satisfactorily addressed my previous comments/queries.

We thank the referee for recognizing the improvements made on our revised manuscript.

Minor point:

Individual data points are not shown for the bar graphs in Figures 3c and 4c as per editorial policy checklist.

Individual data points have been added as requested.

Reviewer #3 (Remarks to the Author):

1. The revised title is almost the same as the original one. The authors need to remove mention of transmembrane domain 6 from the title especially if they think the dimer association process is dynamic and can involve different helices in both the inactive and active states. In addition, GB1 oligomers are a significant part of the manuscript, and the oligomeric interfaces can involve TM1, 4, 5, 6 and 7.

As requested by the referee, we now removed the mention of TM6 in our new proposed title: "*Rearrangement of the transmembrane domain interfaces associated with the activation of a GPCR hetero-oligomer*".

2. The videos need to be removed because there is not sufficient evidence supporting a three-dimensional model of the heterodimer let alone the oligomers, and the authors agree that the videos are an overstatement of their data. The videos also do not reflect the dynamics of receptor conformation.

As requested by the referee, we removed the videos in our newly revised manuscript.

3. Regarding the authors' explanation of persistent GB1-GB2 dimer bands, it is highly unlikely that the presence of a coiled-coil domain would have caused the appearance of these dimer bands under denaturing and reducing conditions. Although coiled-coil interactions are of high affinity, they are still non-covalent, and cannot be maintained once the proteins are denatured. The authors cite a structure paper (Burmakina et al., PNAS, 2014) to show the high affinity coil-coil interaction. However, in that same paper, the western blots of wild-type and various coiled-coil mutants of the receptor only appeared as monomeric bands, indicating that coil-coiled interactions are not sufficient to retain heterodimers once the proteins are denatured. By the same token, the interaction between the ECDs is non-covalent, and is not strong enough to produce the dimer band seen on most gels in this study.

The persistent dimer bands most likely resulted from non-specific crosslinking or non-specific association of the receptor subunits upon denaturation. I think the authors need to indicate this when they explain the limitation of the crosslinking method.

As requested by the first referee, we now include a bar graph representing any quantification of the dimer/total GB1 ratio for each construct analyzed in the basal condition (see new Supp Figure 4). This bar graph illustrates the variability of this ratio from experiment to experiment. As indicated by the referee, this is consistent with a SDS resistant dimer resulting from non-specific crosslinking or non-specific association of the subunits upon denaturation. This is now clearly stated in our manuscript as a drawback of the approach used in our study (see new paragraph page 6 and 7 in the ms). It is the reason why we concentrated our effort in

identifying Cys positions for which a clear CuP-induced crosslinking can be observed, and most importantly on positions for which a change is observed upon receptor activation. Regarding the influence of the coiled coil domain, although we agree that this type of association should not resist to denaturation on SDS buffer, we still think the increased in GB1-GB2 affinity due to this domain likely facilitates the formation of non-specific, SDS resistant dimers. Indeed, in our previous study with mGluR2 that lacks such a coiled coil domain, the relative amounts of dimers are lower than with the GABA_B receptor when the mGlu2 inter-subunit disulfide bound is mutated (Rebuttal Fig. 3). Note that changing the C terminal tail of mGlu2 by that of GB1 (C1) in one subunit, and GB2 (C2) in the other subunit, increases the proportion of dimers detected in the acrylamide gel (Rebuttal Fig. 3).

Rebuttal Fig. 3: Dimer/monomer ratio determined for GABA_B receptors in the present study under basal conditions (“base”), or for the mGlu2 receptor in which the intersubunit disulfide bound is mutated (C121A) carrying the native C terminal tail (Ctr) or in which one subunit contains the C-terminal tail of GB1 (C1) while the other contains that of GB2 (C2). Data with the mGlu2 receptors where those from Xue et al. Nat Chem Biol 2015.

4. On page 9 of the authors’ response, it was indicated that the oligomer band could only be seen when the gels were run with a ratio of 59:1 acrylamide:bisacrylamide, not with a ratio of 29:1 acrylamide:bisacrylamide. Does this mean that the oligomer and heterodimer bands comigrated at a 29:1 acrylamide:bisacrylamide ratio? If this is the case, it would be difficult to determine whether the difference in band intensity was due to changes in the amount of cross-linked heterodimer or oligomer. The authors should consider repeating the gels with 59:1 acrylamide:bisacrylamide to separate the dimer and oligomer bands.

When using the 29:1 acrylamide:bisacrylamide ratio, the large oligomer complexes could not penetrate the gel upon a migration time still allowing the detection of the monomers. As such, any oligomer band is lost when removing the stacking gel. This is why we specifically used a gel with an acrylamide:bisacrylamide ratio of 59:1 to examine the oligomer formation. As illustrated in the rebuttal Fig 4, in which we kept the stacking gel, the referee can notice that the oligomeric band can indeed be detected at the top of the 29:1 gel, then well separated from the dimer band. A direct comparison of the same samples run of both gel types is shown on this figure.

Rebuttal Fig 4: Gel analysis of SNAP-tagged labelled GABA_B receptor constructs before (-) or after (+) CuP treatment. Aliquots from the same samples were run either on a 29:1 acrylamide:bisacrylamide ratio gel (left), or on a 59:1 acrylamide:bisacrylamide ratio gel (right). Note that the oligomer band of the GB1 4.52/6.59 containing receptor obtained after CuP treatment does not penetrate the 29:1 acrylamide:bisacrylamide ratio gel, while it does when using a 59:1 acrylamide:bisacrylamide ratio.

5. Several models for GB1 oligomers have been provided in Figure 7 to explain the different ways three GB1 molecules may interact to form an oligomer. However, the arrangements look quite different from one another, with some potentially precluding the binding of GB2 to at least one GB1 molecule due to steric interaction. These models appear artificial and also suggest that the oligomeric interface is formed in random orientations. Do the authors have any functional data that support the physiological relevance of the various GB1 oligomers?

The schemes added in the revised Fig. 7 were included to help the reader understand what type of crosslinking we are analyzing, as this was not clear to the referee in the first version of our manuscript. The pictograms presented were derived from our dynamic model of the GABA_B oligomer, as illustrated in the **rebuttal Fig. 5** below. Indeed, our view is that the receptor organization is unlikely static, and that the subunits likely move one compare to the other as originally illustrated (schematized) in our movies. The compatibility of the schemes added in Fig. 7 with the GB1-GB2 association is illustrated in **rebuttal Fig. 5**, where the schemes are compared to snap shots of the movie. Of course, these schemes are not intended to represent reality, but to help the reader to understand that we are looking at the crosslinking of three GB1 subunits.

Rebuttal Fig. 5: Schemes used in Fig. 7 to illustrate the possible crosslinking of three GB1 subunits are compared to snapshots of the GABA_B receptor oligomer during its transition from the inactive to the active organization. These schemes illustrate the possible double crosslinking of GB1 subunits through both TM1 and TM4 (top left), TM4 and TM7 (top right), TM4 and TM6 (bottom left) and TM5 and TM1 (bottom right).

Reviewers' Comments:

Reviewer #1:

Remarks to the Author:

The authors have made changes to the manuscript to address many of the outstanding concerns. They have added appropriate language to the manuscript to acknowledge the limitations of their approach.

The only significant concern that remains for this reviewer is the remaining logical inference that an interaction between TM6 domains in some way affects activation. As pointed out in the original review the authors have no evidence that this is the case. If they impose a permanent interaction by crosslinking they can promote activation, but this simply that an interaction between TM6s is "critical for activation", or indeed that any allosteric communication between protomers occurs via TM6 under normal circumstances. The authors (reluctantly) changed the title to remove reference to TM6, but the last two sentences of the abstract still refer to the same idea. These sentences should be removed. This manuscript provides some information regarding how protomers are arranged within GB1-GB2 dimers and oligomers, but not about how interactions between protomers can alter function.

Reviewer #3:

Remarks to the Author:

I think the authors have gone to great length to address the referees' concerns. The paper has also been improved through the two rounds of revision.

Reviewer #1 (Remarks to the Author):

The authors have made changes to the manuscript to address many of the outstanding concerns. They have added appropriate language to the manuscript to acknowledge the limitations of their approach. The only significant concern that remains for this reviewer is the remaining logical inference that an interaction between TM6 domains in some way affects activation. As pointed out in the original review the authors have no evidence that this is the case. If they impose a permanent interaction by crosslinking they can promote activation, but this simply that an interaction between TM6s is "critical for activation", or indeed that any allosteric communication between protomers occurs via TM6 under normal circumstances. The authors (reluctantly) changed the title to remove reference to TM6, but the last two sentences of the abstract still refer to the same idea. These sentences should be removed. This manuscript provides some information regarding how protomers are arranged within GB1-GB2 dimers and oligomers, but not about how interactions between protomers can alter function.

As requested by the referee, we modified our abstract to remain more descriptive. We removed direct interpretation of our data and too strong conclusions. Accordingly, the last sentences have been modified as follow:

"... We bring evidence that agonist activation induces a concerted rearrangement of the various interfaces. While the GB1-GB2 interface is proposed to involve TM5 in the inactive state, cross-linking of TM6s lead to constitutive activity. These data bring insight for our understanding of the allosteric interaction between GPCRs within oligomers."